# YoNoSplat: You Only Need One Model for Feedforward 3D Gaussian Splatting

**Botao Ye**[1,2]  **Boqi Chen**[1,2]  **Haofei Xu**[1]  **Daniel Barath**[1]  **Marc Pollefeys**[1,3]

[1]ETH Zurich   [2]ETH AI Center   [3]Microsoft

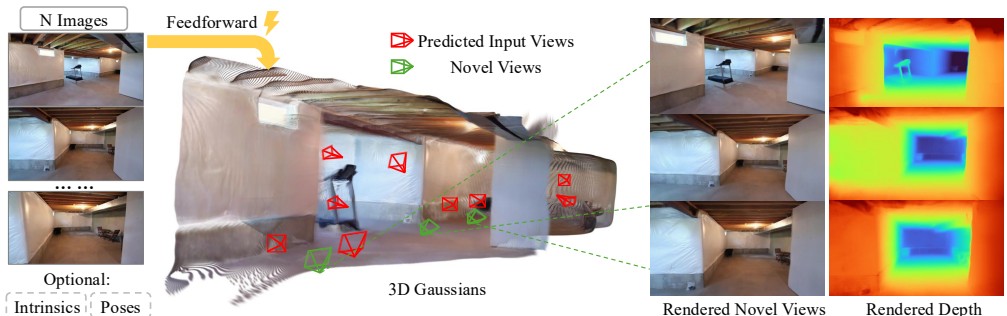

Figure 1: **YoNoSplat**, a versatile feedforward model for rapid 3D reconstruction. Given an arbitrary number of *unposed* and *uncalibrated* multi-view images with wide spatial coverage, it predicts 3D Gaussians and can also utilize ground-truth camera poses or intrinsics when available.

## Abstract

Fast and flexible 3D scene reconstruction from unstructured image collections remains a significant challenge. We present YoNoSplat, a feedforward model that reconstructs high-quality 3D Gaussian Splatting representations from an arbitrary number of multi-view images. Our model is highly versatile, operating effectively with both posed and unposed, calibrated and uncalibrated inputs. YoNoSplat predicts local Gaussians and camera poses for each view, which are aggregated into a global representation using either predicted or provided poses. To overcome the inherent difficulty of jointly learning 3D Gaussians and camera parameters, we introduce a novel mixing training strategy. This approach mitigates the entanglement between the two tasks by initially using ground-truth poses to aggregate local Gaussians and gradually transitioning to a mix of predicted and ground-truth poses, which prevents both training instability and exposure bias. We further resolve the scale ambiguity problem by a novel pairwise camera-distance normalization scheme and by embedding camera intrinsics into the network. Moreover, YoNoSplat also predicts intrinsic parameters, making it feasible for uncalibrated inputs. YoNoSplat demonstrates exceptional efficiency, reconstructing a scene from 100 views (at 280×518 resolution) in just 2.69 seconds on an NVIDIA GH200 GPU. It achieves state-of-the-art performance on standard benchmarks in both pose-free and pose-dependent settings. Our project page is at botaoye.github.io/yonosplat/.

## 1 Introduction

Feedforward Gaussian Splatting (Charatan et al., 2024; Zhang et al., 2025a) has emerged as a promising direction for accelerating 3D scene reconstruction, directly predicting 3D Gaussian parameters (Kerbl et al., 2023) from input images. This approach bypasses the time-consuming per-scene optimization required by methods like NeRF (Mildenhall et al., 2020) and the original 3D Gaussian Splatting (3DGS) (Kerbl et al., 2023). However, the real-world applicability of existing feedforward models is often constrained by restrictive assumptions, such as the need for accurate camera poses (Charatan et al., 2024; Xu et al., 2025), calibrated intrinsics (Ye et al., 2025; Zhang et al., 2025b), or a fixed, limited number of input views (Ye et al., 2025; Hong et al., 2024a).

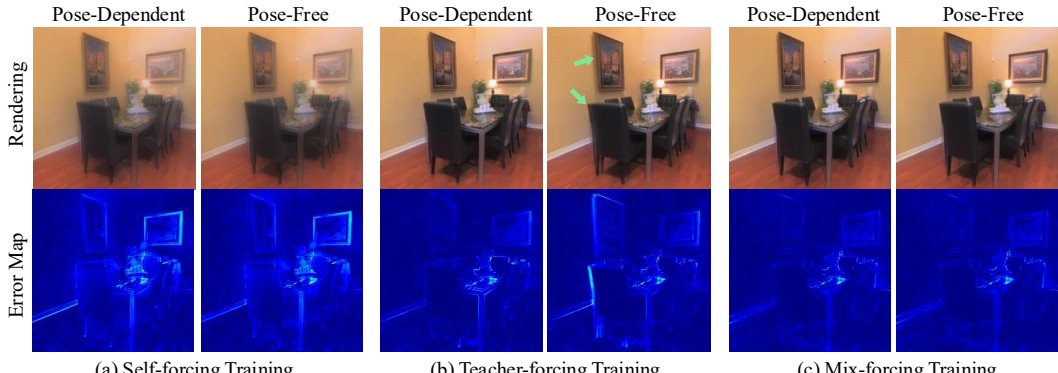

Figure 2: **Effect of different global Gaussian aggregation strategies during training**. (a) Aggregating global Gaussians with predicted camera poses results in poor rendering quality because errors in pose estimation and Gaussian learning compound each other. (b) Using ground-truth poses introduces exposure bias (as indicated by the green arrow: training with ground-truth poses but testing with predicted poses causes misalignment of local Gaussians across different views). (c) Our mix-forcing training achieves high rendering quality in both pose-free and pose-dependent settings.

In practice, scene reconstruction needs to operate under flexible and unconstrained conditions: camera poses may be unavailable or noisy, camera intrinsics unknown, and the number of images may vary significantly. Designing a single model that generalizes across these diverse settings – varying number of views, posed or unposed, calibrated or uncalibrated – remains an open challenge.

In this work, we introduce YoNoSplat, a feedforward model that reconstructs 3D scenes from an arbitrary number of unposed and uncalibrated images, while also seamlessly integrating ground-truth camera information when available. Recent pose-free methods (Ye et al., 2025; Smart et al., 2024) have shown impressive results on sparse inputs (2-4 views) by predicting Gaussians directly into a unified canonical space. However, this approach struggles to scale to large numbers of views. To ensure scalability and versatility, YoNoSplat adopts a different paradigm: it first predicts per-view local Gaussians and their corresponding camera poses, which are then aggregated into a global coordinate system.

This local-to-global design, however, introduces a significant training challenge: the joint learning of camera poses and 3D geometry is highly entangled. Errors in pose estimation can corrupt the learning signal for the Gaussians, and vice-versa. A naive approach that aggregates Gaussians using the model's own predicted poses, known as the self-forcing mechanism (Huang et al., 2025), leads to unstable training and poor performance (Fig. 2a). Conversely, prior methods such as CoPoN-eRF Hong et al. (2024b) adopt a teacher-forcing strategy that relies solely on ground-truth poses for aggregation, which decouples the tasks but also introduces exposure bias (Ranzato et al., 2015). In this case, the model is never trained on its own imperfect pose predictions, causing performance to degrade at inference when it must depend on them (Fig. 2b). To resolve this dilemma, we propose a novel *mix-forcing* training strategy. Training begins with pure teacher-forcing to establish a stable geometric foundation. As training progresses, we gradually introduce the model's predicted poses into the aggregation step. This curriculum balances stability with robustness, enabling YoNoSplat to operate effectively with either ground-truth or predicted poses at test time (Fig. 2c).

A second fundamental challenge is scale ambiguity, which is particularly pronounced when ground-truth depth is unavailable. This ambiguity arises from two sources: training data poses are often defined only up to an arbitrary scale, and jointly estimating intrinsics and extrinsics is an ill-posed problem without a consistent scale reference. Inspired by NoPoSplat (Ye et al., 2025), which highlighted the importance of camera intrinsics for scale recovery, we develop a pipeline that not only uses intrinsic information but also predicts it, enabling reconstruction from uncalibrated images. To address the data-level ambiguity, we systematically evaluate several scene-normalization strategies and find that normalizing by the maximum pairwise camera distance is most effective, as it aligns with the relative pose supervision used during training (Wang et al., 2025c).

Extensive experiments show that our model, even without ground-truth camera inputs, outperforms prior pose-dependent methods, highlighting the strong geometric and appearance priors learned through our training strategy. The approach generalizes across datasets and varying numbers of views, and reconstructs a complete 3D scene from 100 images in just 2.69 seconds.

Our main contributions are as follows:

- We introduce YoNoSplat, the first feedforward model to achieve state-of-the-art performance in both pose-free and pose-dependent settings for an arbitrary number of views.
- We identify the entanglement of pose and geometry learning as a key challenge and propose a mix-forcing training strategy that effectively mitigates training instability and exposure bias.
- We resolve the scale ambiguity problem through an intrinsic-prediction-and-conditioning pipeline and a pairwise distance normalization scheme, enabling reconstruction from uncalibrated images.

## 2 RELATED WORK

**Feedforward 3DGS and NeRF.** Original NeRF (Mildenhall et al., 2020), 3DGS (Kerbl et al., 2023), and their variants (Müller et al., 2022; Barron et al., 2021; Ye et al., 2023) require time-consuming per-scene optimization. To address this inefficiency, numerous feedforward methods (Yu et al., 2021; Hong et al., 2024c; Zhang et al., 2025a; Charatan et al., 2024; Chen et al., 2024) have been proposed. These approaches train neural networks on large-scale datasets to learn geometric and appearance priors, enabling generalization to novel scenes. However, they typically require precise camera poses as input and are restricted to a small number of input views (usually 2–4).

Several works relax individual constraints. For instance, Long-LRM (Ziwen et al., 2025) and Depth-Splat (Xu et al., 2025) reconstruct scenes from multiple input images through a feedforward network, but still rely on accurate camera poses. Recent pose-free methods (Ye et al., 2025; Zhang et al., 2025b) can reconstruct Gaussian-based scenes from unposed images and even outperform pose-dependent counterparts (Charatan et al., 2024; Chen et al., 2024). Yet, they focus primarily on dual-view inputs; while extendable to more views, they remain limited to scenes with sparse coverage. These methods require known intrinsics and operate with a fixed number of views.

In contrast, our work addresses all these challenges simultaneously: we predict local 3D Gaussians, camera intrinsics, and poses feedforward from an arbitrary number of unposed images. The most similar effort is the concurrent AnySplat (Jiang et al., 2025). However, AnySplat cannot leverage available priors such as intrinsics or extrinsics, whereas our method flexibly incorporates them when present. Furthermore, through a carefully designed training paradigm and pose-normalization strategy, YoNoSplat achieves substantially stronger performance.

**Feedforward Point Cloud Prediction.** Another line of work closely related to ours is feedforward point cloud prediction models (Wang et al., 2024; 2025a;c). DUSt3R (Wang et al., 2024) demonstrated that a feedforward model trained on large-scale datasets can accurately predict camera intrinsics and scene geometry without requiring optimization. Subsequent works extended this idea to a larger number of input views (Wang et al., 2025a; Yang et al., 2025; Wang et al., 2025c) and to incremental feedforward reconstruction (Wang et al., 2025b). However, these methods cannot be applied to novel view synthesis due to the discontinuous nature of point clouds. Moreover, they all require ground-truth depth supervision during training. In contrast, by modeling 3D Gaussians as the output representation, YoNoSplat supports both novel view synthesis and effectively utilizes datasets that lack ground-truth depth, such as RealEstate10K Zhou et al. (2018).

## 3 METHOD

We introduce YoNoSplat, a method for the feedforward prediction of 3D Gaussians from multiple images. Our approach supports a wide range of scene scales and can optionally utilize available ground truth camera poses and intrinsics.

**Problem Formulation.** Given $V$ *unposed* images $(\boldsymbol{I}^v)_{v=1}^V$ as input, where $\boldsymbol{I}^v \in \mathbb{R}^{3 \times H \times W}$, our objective is to learn a feedforward network $\boldsymbol{\theta}$ that predicts 3D Gaussians representing the underlying scene. By learning geometric and appearance priors from the training data, our method directly reconstructs new scenes without the need for time-consuming optimization. YoNoSplat do this by first predicting per-view local 3D Gaussians that can be transformed into a global scene representation using the *given* or *predicted* camera poses $\boldsymbol{p}^v$. Specifically, the camera pose parameters are defined as $\boldsymbol{p}^v = [\boldsymbol{R}^v, \boldsymbol{t}^v]$, where $\boldsymbol{R}^v \in \mathbb{R}^{3 \times 3}$ denotes the rotation matrix, $\boldsymbol{t}^v \in \mathbb{R}^3$ represents the translation vector, and $[\cdot]$ indicates the concatenation operation. Furthermore, as described in Sec. 3.2, our network also predicts the camera intrinsics $\boldsymbol{k}^v$, thus can also eliminate the requirement

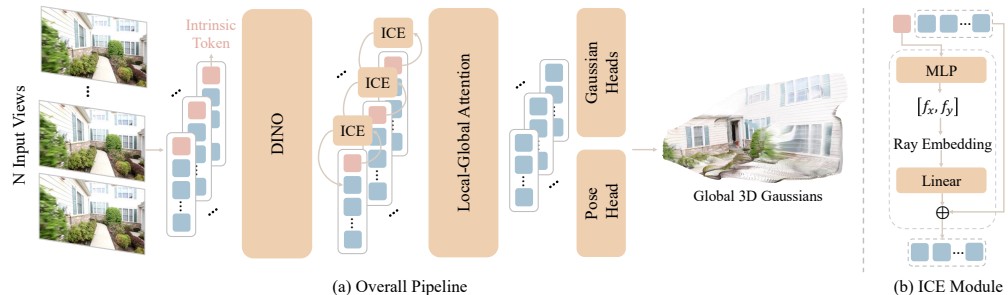

(a) Overall Pipeline  (b) ICE Module

Figure 3: **Overview of YoNoSplat**. (a) Features are extracted with a DINOv2 encoder, followed by local-global attention across images, and finally used to predict camera poses and local 3D Gaussians. (b) The Intrinsic Condition Embedding (ICE) module predicts intrinsic parameters (*i.e.*, focal length), which are then converted into camera rays and re-encoded as conditioning for Gaussian prediction, thereby resolving scale ambiguity.

of camera calibration. Formally, we aim to learn the following mapping:

$$f_{\boldsymbol{\theta}} : \{(\boldsymbol{I}^v)\}_{v=1}^{V} \mapsto \left\{ \cup \left( \boldsymbol{\mu}_j^v, \alpha_j^v, \boldsymbol{r}_j^v, \boldsymbol{s}_j^v, \boldsymbol{c}_j^v \right), \boldsymbol{k}^v, \boldsymbol{p}^v \right\}_{j=1,\dots,H \times W}^{v=1,\dots,V}. \tag{1}$$

Here, $(\boldsymbol{\mu}_j, \alpha_j, \boldsymbol{r}_j, \boldsymbol{s}_j, \boldsymbol{c}_j)$ denote Gaussian parameters (Kerbl et al., 2023), representing the center position, opacity, rotation, scale, and color, respectively. All parameters are initially predicted in the local input camera views and can subsequently be transformed into a global representation using either predicted or given camera poses.

## 3.1 ANALYSIS OF THE GAUSSIAN OUTPUT SPACE AND TRAINING STRATEGY

**Output Space: Local vs. Canonical Prediction.** A fundamental design choice for a feedforward reconstruction model is its output space. Existing methods fall into two main categories. Pose-free models such as NoPoSplat (Ye et al., 2025) and Flare Zhang et al. (2025b) predict Gaussians directly in a unified *canonical space*, which naturally aligns the outputs from all views into a shared coordinate system. In contrast, pose-dependent methods like pixelSplat (Charatan et al., 2024) and MVSplat (Chen et al., 2024) predict Gaussians in a *local*, per-view space and rely on ground-truth camera poses to transform them into the global world frame.

While canonical-space prediction is effective for a small number of views, its performance degrades as the view count increases (see Tab. 7), an observation consistent with findings in related feed-forward point-cloud prediction models (Wang et al., 2025a). To ensure scalability and versatility, YoNoSplat adopts a local prediction paradigm. We architect our model to predict per-view local Gaussians alongside their corresponding camera poses. This design enables our primary goal of pose-free reconstruction by using the predicted poses for aggregation, yet it also retains full compatibility with pose-dependent workflows where ground-truth poses can be supplied. This flexibility is critical for real-world applications, such as map reconstruction, where alignment with a pre-existing, accurate pose distribution is required.

**Training Strategy: Mitigating Pose Entanglement.** Jointly predicting 3D Gaussians and camera parameters is challenging, as errors in one corrupt the other. Using only predicted poses for aggregation (*self-forcing* Huang et al. (2025)) tightly couples the tasks, leading to unstable training and degraded performance (Fig. 2, Tab. 5). Using only ground-truth poses (*teacher-forcing* (Williams & Zipser, 1989)) provides a stable signal but causes *exposure bias* Ranzato et al. (2015), since the model never trains on its own imperfect predictions. To resolve this dilemma, we introduce a novel *mix-forcing* training strategy that combines the benefits of both approaches. Our training curriculum begins by exclusively using ground-truth poses (teacher-forcing) to allow the model to learn a stable geometric foundation. After a predefined number of steps, $t_{\text{start}}$, the probability of using the model's predicted poses for aggregation is linearly increased, eventually reaching a final mixing ratio $r$ at step $t_{\text{end}}$. This strategy effectively mitigates entanglement by first establishing a strong prior for the 3D structure and then gradually adapting the model to both its own predicted distribution and the ground-truth pose distribution, thereby preventing training instability and exposure bias.

## 3.2 MODEL ARCHITECTURE

The overall architecture of YoNoSplat is shown in Fig. 3. We build upon a Vision Transformer (ViT) backbone (Dosovitskiy et al., 2021) and employ a local-global attention mechanism as in

VGGT (Wang et al., 2025a) for robust multi-view feature fusion, which scales more effectively with a large number of input frames than the cross-attention used in prior works (Ye et al., 2025).

**Backbone Network.** Input images $(\boldsymbol{I}^v)_{v=1}^V$ are divided into patches and flattened into tokens. These image tokens are concatenated with a learnable camera intrinsic token and processed by a ViT encoder with a DINOv2 architecture (Oquab et al., 2023). The encoded features then pass through a decoder consisting of $N$ alternating attention blocks (Wang et al., 2025a;c). Each block contains a per-frame self-attention layer for local feature refinement and a global concatenated self-attention layer where tokens from all views are combined to facilitate cross-frame information flow.

**Gaussian Heads.** Following (Ye et al., 2025), we use two separate heads to predict the Gaussian centers and all other parameters. Each head consists of $M$ self-attention layers and a final linear layer. To capture fine-grained detail, we upsample the backbone features by a factor of two before feeding them to the heads and add a skip connection from the input image to combat information loss from the ViT's downsampling.

**Pose Head.** As discussed in Sec. 3.1, YoNoSplat first predicts local Gaussian parameters and then uses either the given or predicted camera poses to transform them into a unified global coordinate system. The camera head consists of an MLP layer, followed by average pooling and another MLP, to predict a 12D camera vector following (Dong et al., 2025; Wang et al., 2025c). This output vector includes the camera translation $\boldsymbol{t}^v$ and a 9D rotation representation (Levinson et al., 2020), which is converted into $\boldsymbol{R}^v$ using SVD orthogonalization. During training, we follow $\pi^3$ Wang et al. (2025c) and supervise the camera pose with a pairwise relative transformation loss (see Sec. 3.4), ensuring that our model remains invariant to the order of input images.

**Intrinsic Head.** Predicting camera poses requires cross-frame information and is thus performed during the decoder stage. In contrast, predicting camera intrinsics can be inferred from individual images. Therefore, we perform intrinsic prediction during the encoder stage. Additionally, conditioning on camera intrinsics helps resolve the scale ambiguity problem, as detailed in Sec. 3.3. Specifically, we concatenate an intrinsic token with the input image tokens, which are then processed by the encoder network, allowing the intrinsic token to aggregate image information. This token is subsequently passed through an MLP layer to predict the camera intrinsics.

## 3.3 RESOLVING SCALE AMBIGUITY

Learning to predict Gaussians from video data encounters a scale ambiguity problem, arising from two main factors: (1) training datasets often provide SfM-derived camera poses that are only defined up to an arbitrary scale, and (2) jointly learning camera intrinsics and extrinsics is an ill-posed problem. We address both factors.

**Scene Normalization.** The ground-truth poses in our training datasets (Zhou et al., 2018; Ling et al., 2024) are obtained using SfM methods (Schonberger & Frahm, 2016), which are only defined up to scale. To avoid scale ambiguity that could hinder learning, it is therefore necessary to normalize the scene during training. Some point-cloud prediction methods (Wang et al., 2024; 2025a) normalize scenes using ground-truth depth, but this is not feasible for datasets without depth annotations.

To address this, we propose and evaluate three normalization strategies:

1. **Max pairwise distance:** given camera centers $\{c_i\}_{i=1}^N$, compute $d_{ij} = \|c_i - c_j\|_2$, and set $s = \max_{i,j} d_{ij}$, then normalize $\hat{c}_i = c_i/s$.
2. **Mean pairwise distance:** use $s = \frac{1}{N(N-1)} \sum_{i \neq j} \|c_i - c_j\|_2$ and normalize as $\hat{c}_i = c_i/s$.
3. **Max translation:** set $s = \max_i \|c_i\|_2$ and normalize $\hat{c}_i = c_i/s$.

As shown in Tab. 6, max pairwise distance normalization performs best and is critical to the model's success. Since we employ relative camera poses, normalizing by the maximum pairwise distance ensures a consistent scale for camera translations during training.

**Intrinsic Condition Embedding (ICE).** As demonstrated in (Ye et al., 2025), camera intrinsic information is crucial for resolving scale ambiguity. However, prior work required ground-truth intrinsics at inference time. To remove this dependency, we introduce our Intrinsic Condition Embedding (ICE) module (Fig. 3b). Specifically, intrinsic parameters are first predicted after the encoder stage using the initial intrinsic token, as detailed in Sec.3.2. Subsequently, to implement intrinsic conditioning, the predicted parameters are transformed into camera rays (Ye et al., 2025), passed through

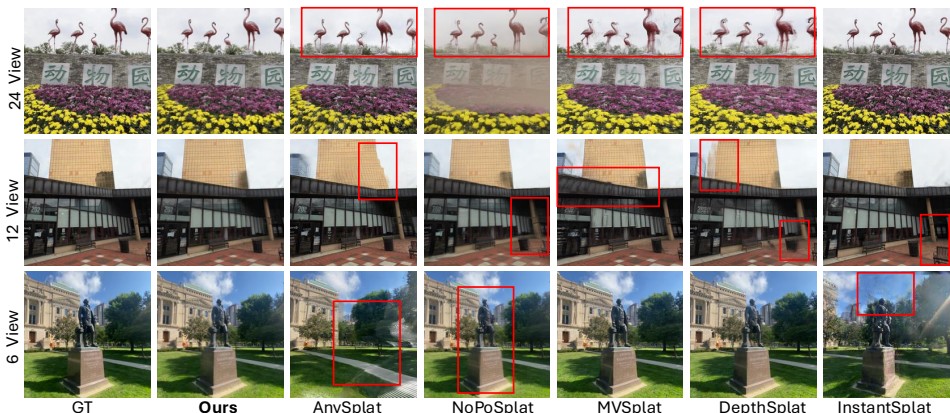

Figure 4: **Qualitative comparison on DL3DV (Ling et al., 2024)**. Here we present our results in the *pose-free, calibration-free* setting, which still produce higher-quality novel view renderings compared to the pose-dependent method DepthSplat (Xu et al., 2025).

a linear layer to obtain embedding features, and then added to the original image features. When ground-truth intrinsics are available, we use them directly to provide more accurate conditioning and thus achieve better performance. In their absence, we instead use the predicted intrinsics for network conditioning. Notably, during training, we condition the network on ground-truth intrinsics rather than the predicted ones. We also experimented with conditioning the decoder on intrinsics predicted by the encoder, but this led to training instability and eventual failure.

### 3.4 MODEL TRAINING

Our models are trained with a multi-task loss as follows:

$$\mathcal{L} = \mathcal{L}_{\text{image}} + \lambda_{\text{intrin}}\mathcal{L}_{\text{intrin}} + \lambda_{\text{pose}}\mathcal{L}_{\text{pose}} + \lambda_{\text{opacity}}\mathcal{L}_{\text{opacity}}. \tag{2}$$

**Rendering Loss.** Following previous works (Chen et al., 2024; Ye et al., 2025), the rendering loss $\mathcal{L}_{\text{image}}$ is set as a linear combination of Mean Squared Error (MSE) and LPIPS (Zhang et al., 2018) loss, which is employed to optimize the Gaussians. During training, we randomly sample 4 target views and render the corresponding images using their ground truth camera poses, and then the rendered images are compared against the ground truth image.

**Intrinsic Loss.** The intrinsic loss $\mathcal{L}_{\text{intrin}}$ is used to train the intrinsic prediction head, which is the $l_2$ distance between the predicted focal length with the ground truth.

**Pose Loss.** Following Wang et al. (2025c), we supervise the pose prediction head with relative pose loss. Specifically, for each pair of input views $(i, j)$ and their predicted pose $(\hat{\mathbf{p}}_i, \hat{\mathbf{p}}_j)$, we first calculate their relative pose $\hat{\mathbf{p}}_{i\leftarrow j} = \hat{\mathbf{p}}_i^{-1}\hat{\mathbf{p}}_j$. The loss is then calculated as $\mathcal{L}_{\text{pose}} = \frac{1}{N(N-1)}\sum_{i\neq j}(\mathcal{L}_{\text{R}}(i,j) + \lambda_t\mathcal{L}_t(i,j))$, where $N$ is the number of input views. Following Dong et al. (2025); Wang et al. (2025c), the rotation $\mathcal{L}_{\text{R}}$ and translation losses $\mathcal{L}_t$ are calculated as:

$$\mathcal{L}_{\text{R}}(i,j) = \arccos((\text{tr}\left((\mathbf{R}_{i\leftarrow j})^\top\hat{\mathbf{R}}_{i\leftarrow j}\right) - 1)/2), \mathcal{L}_t(i,j) = \mathcal{H}_\delta(\hat{\mathbf{t}}_{i\leftarrow j} - \mathbf{t}_{i\leftarrow j}). \tag{3}$$

Here, $\text{tr}(\cdot)$ denotes the trace of a matrix, and $\mathcal{H}_\delta(\cdot)$ calculate the Huber loss.

**Opacity Loss.** Since a Gaussian is predicted per pixel, the total number of Gaussians grows rapidly as the number of views increases. To mitigate this issue, we apply an opacity regularization loss following Ziwen et al. (2025) to promote sparsity. Specifically, $\mathcal{L}_{\text{opacity}} = \frac{1}{M}\sum_{i=1}^{M}|o_i|$, where $M$ is the total number of Gaussians. We then prune those with $o_i < 0.005$. We observed that this removes around $20\% - 70\%$ of the Gaussians, depending on the number of images and their overlap.

### 3.5 EVALUATION

For evaluation in the pose-dependent setting, we render the target view using the corresponding ground-truth camera poses. In contrast, under the pose-free setting, the predicted camera space may differ from the ground-truth poses obtained via SfM methods. To faithfully assess the quality of Gaussian reconstruction, we follow prior pose-free approaches (Ye et al., 2025; Wang et al., 2021;

Table 1: **Novel view synthesis comparison under various input settings.** We report results on DL3DV (Ling et al., 2024) with 6, 12, and 24 input views, where $p$, $k$, and Opt denote using ground-truth poses, intrinsics, and post-optimization. Our method consistently outperforms previous SOTA approaches, including the pose-dependent DepthSplat, even without prior information.

| Method | $p$ | $k$ | Opt | 6v | | | 12v | | | 24v | | |
|---|---|---|---|---|---|---|---|---|---|---|---|---|
| | | | | PSNR ↑ | SSIM ↑ | LPIPS ↓ | PSNR ↑ | SSIM ↑ | LPIPS ↓ | PSNR ↑ | SSIM ↑ | LPIPS ↓ |
| MVSplat | ✓ | ✓ | | 22.659 | 0.760 | 0.173 | 21.289 | 0.709 | 0.224 | 19.975 | 0.662 | 0.269 |
| DepthSplat | ✓ | ✓ | | 23.418 | 0.797 | **0.136** | 21.911 | 0.753 | 0.179 | 20.088 | 0.690 | 0.240 |
| **Ours** | ✓ | ✓ | | **24.717** | **0.817** | 0.139 | **23.285** | **0.773** | **0.177** | **22.664** | **0.758** | **0.192** |
| NoPoSplat | | ✓ | | 22.766 | 0.743 | 0.179 | 19.380 | 0.563 | 0.318 | 17.860 | 0.495 | 0.397 |
| **Ours** | | ✓ | | **24.887** | **0.819** | **0.138** | **23.149** | **0.758** | **0.183** | **22.354** | **0.731** | **0.205** |
| AnySplat | | | | 19.027 | 0.554 | 0.235 | 18.940 | 0.549 | 0.262 | 19.703 | 0.596 | 0.249 |
| **Ours** | | | | **24.531** | **0.804** | **0.142** | **22.933** | **0.746** | **0.187** | **22.174** | **0.720** | **0.209** |
| InstantSplat | | | ✓ | 21.677 | 0.627 | 0.273 | 20.792 | 0.580 | 0.316 | 18.493 | 0.510 | 0.381 |
| **Ours** | | | ✓ | **27.533** | **0.866** | **0.106** | **26.126** | **0.820** | **0.133** | **25.855** | **0.814** | **0.136** |

Fan et al., 2024), which first predict the target camera poses and then render the images using these predicted poses for evaluation. The prediction of target camera poses follows (Ye et al., 2025), which optimizes the poses through a photometric loss based on the predicted 3D Gaussians.

**Optional Post-Optimization.** After YoNoSplat predicts 3D Gaussians and pose parameters, we optionally perform a fast post-optimization. Specifically, we optimize the predicted camera poses along with the Gaussian centers and colors, while keeping all other parameters fixed (see the appendix for details). The results in Tab. 1 show that this optional optimization can further improve performance with a reasonable time cost.

# 4 EXPERIMENTS

## 4.1 EXPERIMENTAL SETUP

**Datasets.** We train on RealEstate10K (RE10K) (Zhou et al., 2018) and DL3DV (Liu et al., 2021) using the official splits. RE10K consists of indoor real-estate videos (67,477 train / 7,289 test). DL3DV (Ling et al., 2024) contains 10,000 outdoor videos, 140 for testing. For evaluation on RE10K, we keep test sequences with $\geq 200$ frames (1,580 sequences) and use 6 context views due to the smaller scene scale. On DL3DV, we test with $(6, 12, 24)$ input views and maximum frame gaps $(50, 100, 150)$. For generalization, we evaluate the DL3DV-trained model on ScanNet++ (Yesh-wanth et al., 2023) by sampling $(32, 64, 128)$ views per sequence with a fixed target view. Inputs are selected by farthest point sampling over camera centers; 8 views are randomly held out as validation.

**Implementation Details.** YoNoSplat is implemented using PyTorch. The encoder employs the DINOv2 Large model (Oquab et al., 2023) with 24 attention layers, and the decoder consists of 18 alternating-attention layers. The parameters of the backbone, Gaussian center head, and camera pose head are initialized from $\pi^3$ (Wang et al., 2025c), while the remaining layers are initialized randomly. During training, we randomly select the number of input views between 2 and 32 views and sample 4 target views. We train models at two different resolutions, $224 \times 224$ and $280 \times 518$. The $224 \times 224$ model is trained on 16 GH200 GPUs for 150k steps with a batch size of 2 for each, while the $280 \times 518$ model is initialized from the pretrained $224 \times 224$ weights and further trained on 32 GH200 GPUs for another 150k steps with a batch size of 1.

**Evaluation Metrics.** For the novel view synthesis task, we evaluate with the commonly used metrics: PSNR, SSIM, and LPIPS. For pose estimation, we report the area under the cumulative angular pose error curve (AUC) thresholded at 5°, 10°, and 20° (Sarlin et al., 2020; Edstedt et al., 2024).

**Baselines.** We compare against SOTA representative sparse-view generalizable methods on novel view synthesis: 1) *Optimization-based*: InstantSplat (Fan et al., 2024); 2) *Pose-dependent*: MVS-plat (Chen et al., 2024), DepthSplat (Xu et al., 2025); 3) *Pose-free*: NoPoSplat (Ye et al., 2025) and AnySplat (Jiang et al., 2025). For relative pose estimation, we compare against SOTA methods: MASt3R (Leroy et al., 2024), VGGT (Wang et al., 2025a), and $\pi^3$ (Wang et al., 2025c).

## 4.2 EXPERIMENTAL RESULTS AND ANALYSIS

**Novel View Synthesis.** To evaluate our method on complex real-world scenes, we test it on DL3DV with varying numbers of input views, scene scales, and input priors. As shown in Table 1, our model consistently outperforms previous SOTA approaches. Notably, YoNoSplat surpasses leading

| Method | $p$ | $k$ | PSNR ↑ | SSIM ↑ | LPIPS ↓ |
|---|---|---|---|---|---|
| DepthSplat | ✓ | ✓ | 24.156 | 0.846 | 0.145 |
| NoPoSplat | | ✓ | 22.175 | 0.750 | 0.207 |
| **Ours** | ✓ | ✓ | 25.037 | 0.848 | 0.134 |
| **Ours** | | ✓ | **25.395** | **0.857** | **0.131** |
| **Ours** | | | 24.571 | 0.823 | 0.144 |

Table 2: **NVS comparison on the RE10k dataset (6 input views) under different prior settings.** Our model consistently achieves the best performance.

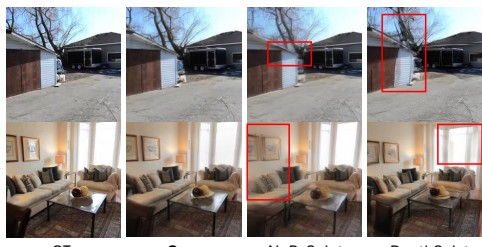
GT **Ours** NoPoSplat DepthSplat

Figure 5: **Qualitative comparison on RealEstate10K (Zhou et al., 2018).** Our *pose-free, calibration-free* method enables a more coherent fusion of multi-view contents.

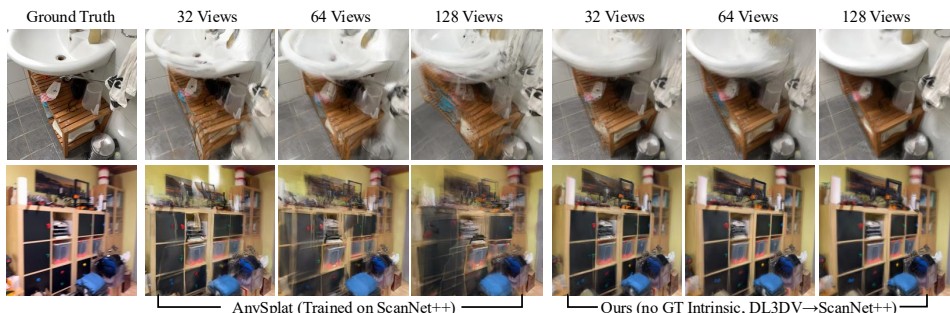

Figure 6: **Qualitative comparison on ScanNet++.** YoNoSplat generalizes well to ScanNet++ and demonstrates more coherent fusion of Gaussians across different views compared to AnySplat. Moreover, adding more inputs leads to better rendering quality, as more information is provided.

pose-free methods like NoPoSplat and AnySplat by a substantial margin. More strikingly, even in the most challenging *pose-free, intrinsic-free* setting, our model outperforms the SOTA *pose-dependent* method, DepthSplat, across all view counts. This highlights our model's ability to learn powerful priors that compensate for the lack of ground-truth camera information. Qualitatively, Fig. 4 shows that our reconstructions have better cross-view consistency, avoiding the artifacts and inaccurate geometry seen in baselines. The results in Table 1 also reveal that as the number of input views and scene scale increase (e.g., 12 and 24 views), providing ground-truth extrinsics can further improve performance, acknowledging the inherent difficulty of pose estimation in large-scale environments. Furthermore, a fast, optional post-optimization of the predicted Gaussians and poses yields additional performance gains. On the indoor RealEstate10K dataset (Table 2), our 6-view model continues this trend, outperforming both pose-free and pose-dependent SOTA methods, as shown qualitatively in Fig. 5.

We recommend that readers watch our *supplementary videos* for more results.

**Cross-Dataset Generalization.** To assess generalizability, we train YoNoSplat on the DL3DV dataset and evaluate it on ScanNet++ without fine-tuning. We compare against AnySplat, which is trained on ScanNet++. As shown in Tab, 3, our model significantly outperforms this baseline across all metrics and view counts, despite AnySplat's training-domain advantage. It is worth noting that our performance consistently improves as more input views are provided, demonstrating our model's ability to effectively integrate additional information. The qualitative results in Fig. 6 corroborate this, showing YoNoSplat produces significantly sharper and more coherent reconstructions, demonstrating a better fusion of information across different views. In contrast, the renderings from AnySplat appear blurrier and contain more noticeable artifacts. These findings highlight our model's robust ability to generalize to novel datasets not seen during training.

**Camera Pose Estimation.** As shown in Tab. 4, our model with small resolution input ($224 \times 224$) already achieves the best performance compared to state-of-the-art methods, while our model with large input resolution ($518 \times 280$) obtains the best performance compared to other state-of-the-art approaches. Moreover, we evaluate our model trained on DL3DV but tested on RealEstate10K (indicated as DL3DV→RE10K in Tab. 4), ensuring that none of the methods are trained on the RealEstate10K dataset. The results demonstrate that our method generalizes well and outperforms all baselines, highlighting that training with a rendering loss also benefits pose estimation.

Table 3: **Generalization to ScanNet++.** Trained on DL3DV and tested on ScanNet++, our model outperforms AnySplat, despite AnySplat being trained on ScanNet++. Input images are sampled from full sequences with a fixed target view, and our performance improves with more input views.

| Method | 32v | | | 64v | | | 128v | | |
|---|---|---|---|---|---|---|---|---|---|
| | PSNR ↑ | SSIM ↑ | LPIPS ↓ | PSNR ↑ | SSIM ↑ | LPIPS ↓ | PSNR ↑ | SSIM ↑ | LPIPS ↓ |
| AnySplat | 14.054 | 0.494 | 0.468 | 15.982 | 0.551 | 0.412 | 16.988 | 0.583 | 0.386 |
| **Ours w/o GT** $k$ | 16.886 | 0.600 | 0.432 | 17.368 | 0.608 | 0.413 | 17.641 | 0.617 | 0.405 |
| **Ours w/ GT** $k$ | **17.935** | **0.659** | **0.380** | **18.833** | **0.688** | **0.342** | **19.284** | **0.701** | **0.325** |

Table 4: **Pose estimation comparison.** Our method achieves the best pose estimation with a smaller input resolution ($224 \times 224$) and further improves with a larger resolution ($518 \times 280$). We also report zero-shot results on RE10k using a model trained exclusively on DL3DV (so that none of the models are trained on RE10k); our method still outperforms all others.

| Method | DL3DV | | | RealEstate10K | | |
|---|---|---|---|---|---|---|
| | 5° ↑ | 10° ↑ | 20° ↑ | 5° ↑ | 10° ↑ | 20° ↑ |
| MASt3R $_{518 \times 288}$ | 0.778 | 0.883 | 0.941 | 0.609 | 0.776 | 0.878 |
| NoPoSplat$_{256 \times 256}$ | 0.538 | 0.735 | 0.853 | 0.443 | 0.627 | 0.755 |
| VGGT $_{518 \times 280}$ | 0.700 | 0.848 | 0.924 | 0.566 | 0.753 | 0.867 |
| $\pi^3$ $_{518 \times 280}$ | 0.795 | 0.897 | 0.949 | 0.705 | 0.841 | 0.916 |
| **Ours**$_{224 \times 224}$ | 0.833 | 0.917 | 0.958 | 0.722 | 0.852 | 0.923 |
| **Ours**$_{224 \times 224}$ **(DL3DV→RE10K)** | | - | | 0.74 | 0.859 | 0.924 |
| **Ours**$_{518 \times 280}$ | **0.844** | **0.922** | **0.961** | **0.813** | **0.904** | **0.951** |
| **Ours**$_{518 \times 280}$ **(DL3DV→RE10K)** | | - | | 0.78 | 0.884 | 0.939 |

Table 5: **Mix-forcing** achieves the best balance of pose-free and pose-dependent performance.

| Method | Pose-dependent | | | Pose-free | | |
|---|---|---|---|---|---|---|
| | PSNR | SSIM | LPIPS | PSNR | SSIM | LPIPS |
| Mix-forcing | 25.212 | 0.848 | 0.133 | **25.587** | **0.854** | **0.130** |
| Self-forcing | 24.150 | 0.815 | 0.150 | 24.652 | 0.831 | 0.145 |
| Teacher-forcing | **25.228** | **0.850** | **0.131** | 25.300 | 0.851 | 0.131 |

Table 6: **Pose normalization.** Max pairwise distance normalization leads to best performance.

| Norm. | PSNR ↑ | SSIM ↑ | LPIPS ↓ |
|---|---|---|---|
| $\max_{i,j} d_{ij}$ | **25.212** | **0.848** | **0.133** |
| $\mathrm{mean}_{i,j} d_{ij}$ | 24.950 | 0.845 | 0.135 |
| $\max_i d_i$ | 22.739 | 0.756 | 0.184 |
| No Norm. | 22.662 | 0.757 | 0.185 |

## 4.3 Ablation Studies

For this ablation, we train the model with only 6 input views for faster training, without compromising generalizability. As a result, the performance is slightly better compared with Tab. 5.

**Effectiveness of the Mix-Forcing Strategy.** We compare *mix-forcing* with pure *teacher-forcing* (ground-truth poses) and *self-forcing* (predicted poses). As shown in Table 5, self-forcing performs worst in both settings, confirming that entangled pose–geometry learning causes instability. Teacher-forcing excels with ground-truth poses but drops under pose-free evaluation due to exposure bias. Mix-forcing balances these trade-offs, achieving the best pose-free results while remaining competitive in the pose-dependent case, yielding a more robust and versatile model.

**Importance of Scene Normalization.** As discussed in Sec. 3.3, scene normalization is essential for training on datasets with poses that are only defined up-to-scale. Tab. 6 demonstrates this empirically. Without any normalization, the model's performance is severely degraded. We compare our chosen strategy, normalizing by the maximum pairwise distance between camera centers, against two alternatives: normalizing by the mean pairwise distance and by the maximum camera translation from the origin. The results clearly indicate that max pairwise distance normalization yields the best performance. This is because it provides a consistent and robust scale reference for camera translations that aligns directly with the relative pose supervision loss used during training.

## 5 Conclusion

In this work, we introduced YoNoSplat, a versatile feedforward model for high-quality 3D Gaussian reconstruction from an arbitrary number of images, uniquely capable of operating in both pose-free/pose-dependent and calibrated/uncalibrated settings. We address two key challenges: the entanglement of geometry and pose learning, and scale ambiguity. Our novel *mix-forcing* training strategy resolves the former by balancing training stability and mitigating exposure bias. For the latter, we combine a robust *max pairwise distance normalization* with an *Intrinsic Condition Embedding (ICE)* module that enables reconstruction from uncalibrated inputs. These contributions significantly advance the flexibility and robustness of feedforward 3D reconstruction.

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

## A    MORE IMPLEMENTATION DETAILS

**Training.** During the training process, we first randomly sample a clip from the training videos, the context image are sampled with farthest point sampling based on camera centers from training video clips to ensure sufficient input coverage. The target images are randomly sampled from the whole video clip. We employ the AdamW optimizer (Loshchilov & Hutter, 2018), setting the initial learning rate for the backbone to $2 \times 10^{-5}$ and other parameters to $2 \times 10^{-4}$. The weight of intrinsic loss $\lambda_{\text{intrin}}$, pose loss $\lambda_{\text{pose}}$, and opacity loss are set to 0.5, 0.1, and 0.01 respectively. For the mix-forcing training, we set $t_{start} = 80k$, $t_{end} = 100k$, and mixing ratio $r = 0.1$. Moreover, during the training process, we skip the batches if the pose loss is larger than 1 to avoid abnormal input sequences affect the training stability.

**Evaluation.** For comparison with baseline methods on novel view synthesis, we use the $224 \times 224$ version of our model to ensure a fair comparison, as it best aligns with the experimental settings of the other baselines. Different prior methods adopt different input resolutions (*e.g.*, MVSplat (Chen et al., 2024) and NoPoSplat (Ye et al., 2025) use $256 \times 256$, while DepthSplat (Xu et al., 2025) uses $256 \times 448$). Due to computational constraints and to avoid noise from in-house reproduction, it is not feasible to retrain all baselines and our model at a unified resolution. However, we have taken care to ensure that the comparisons remain fair and meaningful: 1) Our model has the smallest receptive size among all compared methods. Since all methods first center-crop and then resize, square crops result in minimal receptive coverage. We use this model for novel view synthesis comparisons to maintain fairness. 2) Because our model has the smallest receptive size, we can center-crop and resize the rendered outputs of other methods, ensuring that all comparisons are performed on the same image content.

**Optional Post-Optimization.** This fast optimization step refines the predicted camera poses, Gaussian centers, and colors for 200 iterations. We use learning rates of 0.005 for pose parameters, 0.0016 for Gaussian means, and 0.0025 for colors. The total optimization time varies with the number of input views: 17.7s for 6 views, 51.1s for 12 views, and 165s for 24 views.

## B    MORE EXPERIMENTAL ANALYSIS

### B.1    ON THE UTILITY OF GROUND-TRUTH POSE PRIORS

A noteworthy and somewhat counter-intuitive result emerges from our experiments, as shown in Tables 1 and 2. In settings with a small number of input views (*e.g.*, 6 views), our model operating in the fully *pose-free* setting outperforms its *pose-dependent* counterpart, which is supplied with ground-truth camera poses. We hypothesize that this is due to the inherent noise and potential inconsistencies within the "ground-truth" poses themselves, which are typically derived from Structure-from-Motion (SfM) pipelines. For sparse-view reconstructions, minor inaccuracies in SfM poses can lead to subtle misalignments when aggregating local Gaussians. In contrast, our pose-free model is optimized end-to-end to produce a set of camera poses and a 3D representation that are maximally photometrically consistent with each other. This internal self-consistency can lead to higher-quality renderings than forcing the model to align with a slightly imperfect ground-truth coordinate system. Moreover, the slight misalignment of the target pose also contributes to this.

However, this trend reverses as the number of input views and the scene scale increase (see Tab. 1). For larger view counts (*e.g.*, 12 and 24), the pose-dependent setting regains its advantage. This occurs because pose estimation becomes more challenging as the scene scale increases, whereas SfM-based ground-truth poses provide a strong geometric prior. This analysis highlights the robustness of our model: it can learn priors strong enough to compensate for noisy ground-truth data in sparse-view scenarios, while also effectively leveraging ground-truth pose information when the scene scale is large.

### B.2    ABLATION ON OUTPUT GAUSSIAN SPACE

As discussed in Sec. 3.1, a fundamental design choice is the output representation space. We compare our approach of predicting Gaussians in a **local**, per-view space against the alternative of predicting them directly into a unified **canonical** space. Tab. 7 shows that the local prediction strategy

Table 7: **Comparison of Gaussian representations.** Local Gaussian performs better on the 6-view setting.

| Representation | PSNR ↑ | SSIM ↑ | LPIPS ↓ |
|---|---|---|---|
| Local Gaussian | **25.587** | **0.854** | **0.130** |
| Canonical Gaussian | 24.104 | 0.819 | 0.172 |

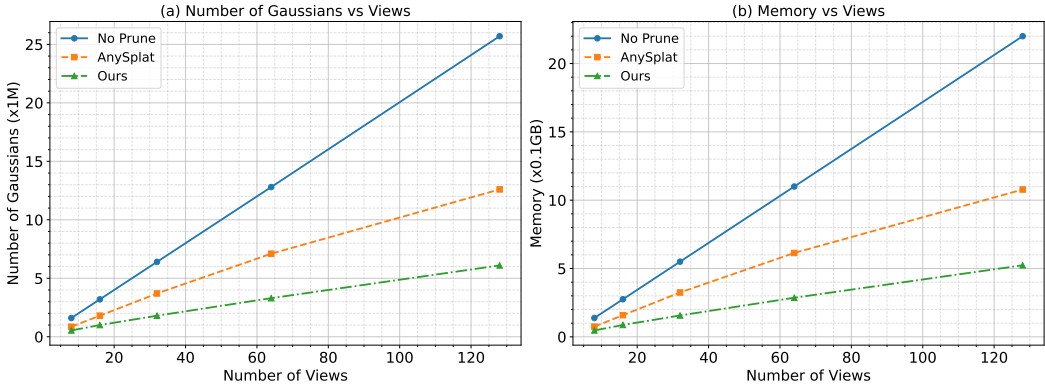

Figure 7: **Scalability with respect to the number of input views.** (a) Average number of Gaussians and (b) memory usage on the ScanNet++ evaluation set when varying the number of input views. "No Prune" denotes vanilla per-pixel Gaussian prediction without any pruning, "AnySplat" Jiang et al. (2025) uses voxel-based pruning, and **Ours** uses opacity-based pruning.

significantly outperforms the canonical one on all metrics. This result empirically validates our hypothesis that predicting in a local space is more scalable and robust, especially as the number of views increases, as it avoids the difficulty of forcing a single network to align features from multiple views into one arbitrary coordinate frame.

### B.3 IMPACT OF INTRINSIC CONDITION EMBEDDING (ICE)

Table 8: **Effect of ICE Module.** Using the intrinsics predicted by our model leads to better performance compared to training without intrinsic conditioning.

| | PSNR ↑ | SSIM ↑ | LPIPS ↓ |
|---|---|---|---|
| GT Intrinsic | 25.587 | 0.854 | 0.130 |
| Pred Intrinsic | 24.711 | 0.825 | 0.141 |
| No Intrinsic | 24.481 | 0.813 | 0.149 |

Table 8 evaluates three inference scenarios: (1) ground-truth intrinsics, (2) predicted intrinsics, and (3) no intrinsic conditioning. Removing intrinsics causes a clear performance drop, confirming their importance for resolving scale ambiguity. Using predicted intrinsics significantly outperforms the no-intrinsic baseline and comes close to ground-truth, demonstrating that ICE enables high-quality reconstruction even from uncalibrated inputs.

### B.4 SCALABILITY IN GAUSSIAN NUMBER AND MEMORY

To compare the efficiency of our opacity-based pruning with vanilla per-pixel Gaussian prediction and AnySplat, we evaluate all methods on the ScanNet++ dataset while varying the number of input views. For each setting, we report the average number of Gaussians and the corresponding memory consumption over all scenes in the evaluation set. As shown in Fig. 7, the "No Prune" baseline exhibits a rapid growth in both Gaussian count and memory as the number of views increases. AnySplat mitigates this growth using voxel-based pruning, but still requires a large number

Table 9: **Sparse-view novel view synthesis on RealEstate10K using only 2 or 3 input views.** Our method matches or surpasses prior approaches with 2 views and achieves a clear margin over NoPoSplat when 3 views are available.

| Method | 2 view | | | 3 view | | |
|---|---|---|---|---|---|---|
| | PSNR↑ | SSIM↑ | LPIPS↓ | PSNR↑ | SSIM↑ | LPIPS↓ |
| pixelNeRF | 19.824 | 0.626 | 0.485 | - | - | - |
| pixelSplat | 23.848 | 0.806 | 0.185 | - | - | - |
| MVSplat | 23.977 | 0.811 | 0.176 | - | - | - |
| NoPoSplat | **25.033** | **0.838** | 0.160 | 26.619 | 0.872 | 0.127 |
| Ours | 24.917 | 0.834 | **0.154** | **27.528** | **0.892** | **0.106** |

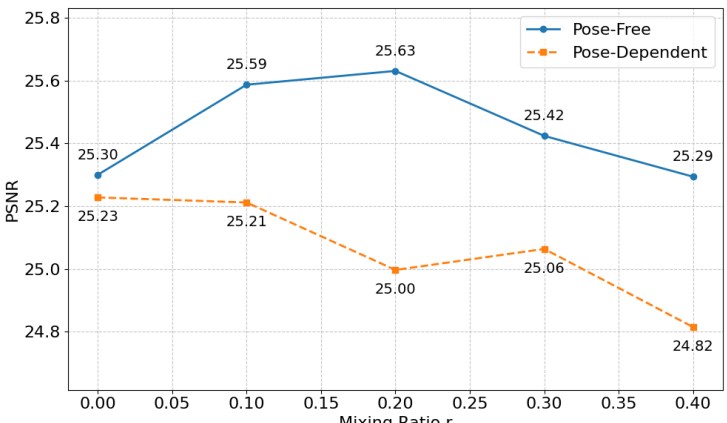

Figure 8: **Effect of the mixing ratio** $r$**.** PSNR on the RealEstate10K dataset with 6 input views under pose-free and pose-dependent settings as a function of the training hyperparameter $r$. We choose $r = 0.1$ as it provides a good trade-off, improving pose-free performance while keeping pose-dependent performance nearly unchanged.

of Gaussians. In contrast, our opacity-based pruning strategy significantly reduces the number of Gaussians and memory usage across all view counts, achieving better scalability than both baselines while preserving reconstruction quality.

### B.5 SPARSE-VIEW NOVEL VIEW SYNTHESIS

To assess robustness under extremely sparse observations, we evaluate YoNoSplat on RealEstate10K using only 2 or 3 input views. As reported in Tab. 9, our model remains strong in this sparse-view setting. With only 2 views, YoNoSplat attains comparable PSNR/SSIM to the previous state of the art, NoPoSplat, while achieving a lower LPIPS (0.154 vs. 0.160), indicating better perceptual quality. When 3 views are available, YoNoSplat clearly outperforms NoPoSplat across all metrics.

### B.6 ABLATION ON THE MIX RATIO

We evaluate novel view synthesis performance (PSNR) under both pose-free and pose-dependent settings across different values of the mixing ratio $r$. Results in Fig. 8 show that larger values of $r$ (e.g., $r = 0.2$) further improve pose-free performance but substantially degrade pose-dependent performance. We therefore choose $r = 0.1$, which provides a good trade-off: pose-dependent performance remains nearly identical to that of $r = 0$, while pose-free performance improves noticeably. Overall, pose-free performance consistently increases for moderate values of $r$ ($r < 0.3$), whereas pose-dependent performance is stable around $r = 0.1$ and gradually decreases as $r$ increases.

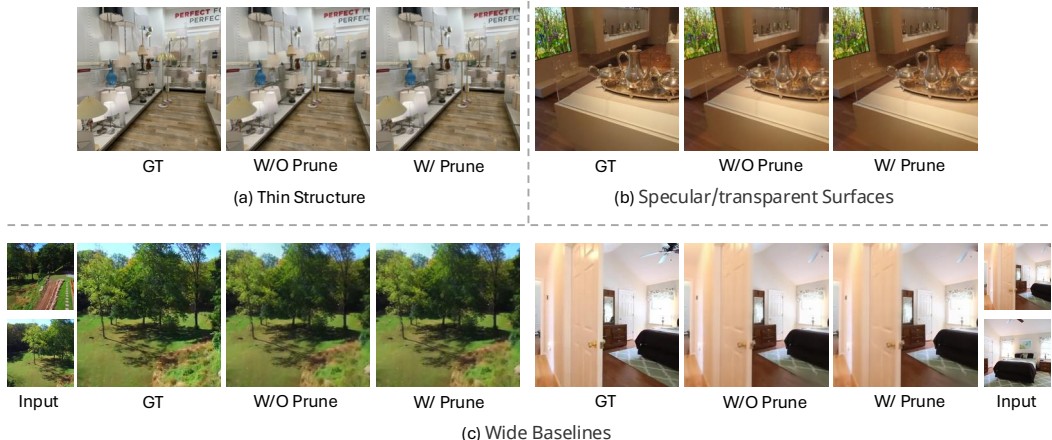

Figure 9: **Effect of Gaussian pruning under challenging conditions.** We compare models without pruning and with pruning applied on (a) thin structures, (b) specular and transparent surfaces, and (c) wide-baseline views. In all cases, pruning does not introduce visible degradation, and the results remain visually consistent with the ground truth.

Table 10: **Effect of noise in camera intrinsics on novel view synthesis.** We perturb the intrinsic parameters by different relative noise levels and report performance with and without intrinsic conditioning.

| Metric | No GT Intrinsic | 0% | 5% | 10% | 15% | 20% |
|---|---|---|---|---|---|---|
| PSNR ↑ | 24.711 | **25.587** | 24.913 | 23.681 | 22.646 | 21.942 |
| SSIM ↑ | 0.825 | **0.854** | 0.834 | 0.786 | 0.742 | 0.712 |
| LPIPS ↓ | 0.141 | **0.130** | 0.138 | 0.157 | 0.179 | 0.199 |

## B.7 ANALYSIS OF GAUSSIAN PRUNING

We analyze the impact of our Gaussian pruning strategy on reconstruction quality across challenging visual conditions, including thin structures, specular or transparent surfaces, and wide-baseline viewpoints. These scenarios are known to be sensitive to over-pruning, as removing geometrically important but weakly activated Gaussians may potentially degrade fine details or appearance consistency. Fig. 9 shows qualitative comparisons between novel views without pruning and those with pruning applied, along with the ground truth. We observe that the rendering quality remains visually indistinguishable after pruning in all tested cases. In particular, fine geometric structures (*e.g.*, lamp stands), reflective or transparent objects (*e.g.*, glassware), and scenes with extremely wide camera baselines are preserved without noticeable artifacts or loss of details. Quantitatively, we prune Gaussians whose opacity satisfies $o_i < 0.005$. After pruning, the average PSNR drop over the entire evaluation set is less than $0.01$ dB compared to the unpruned model. This confirms that the removed Gaussians contribute negligibly to the final rendering and that our pruning strategy does not sacrifice reconstruction accuracy, even under difficult imaging conditions. Overall, this study demonstrates that our Gaussian pruning method is both safe and effective: it significantly reduces the Gaussian number while preserving high-fidelity reconstruction quality.

## B.8 EFFECT OF INTRINSIC NOISE ON CONDITIONING

We first investigate how noisy camera intrinsics affect the benefit of our intrinsic conditioning. As shown in Tab. 10, small perturbations (e.g., $5\%$ noise) are handled well: the model with noisy intrinsics still outperforms the *No GT Intrinsic* baseline across all metrics, and remains close to the clean-$0\%$ setting. When the noise level becomes larger ($\geq 10\%$), performance naturally degrades, since incorrect physical cues misguide the reconstruction. Nevertheless, these noise levels are sub-

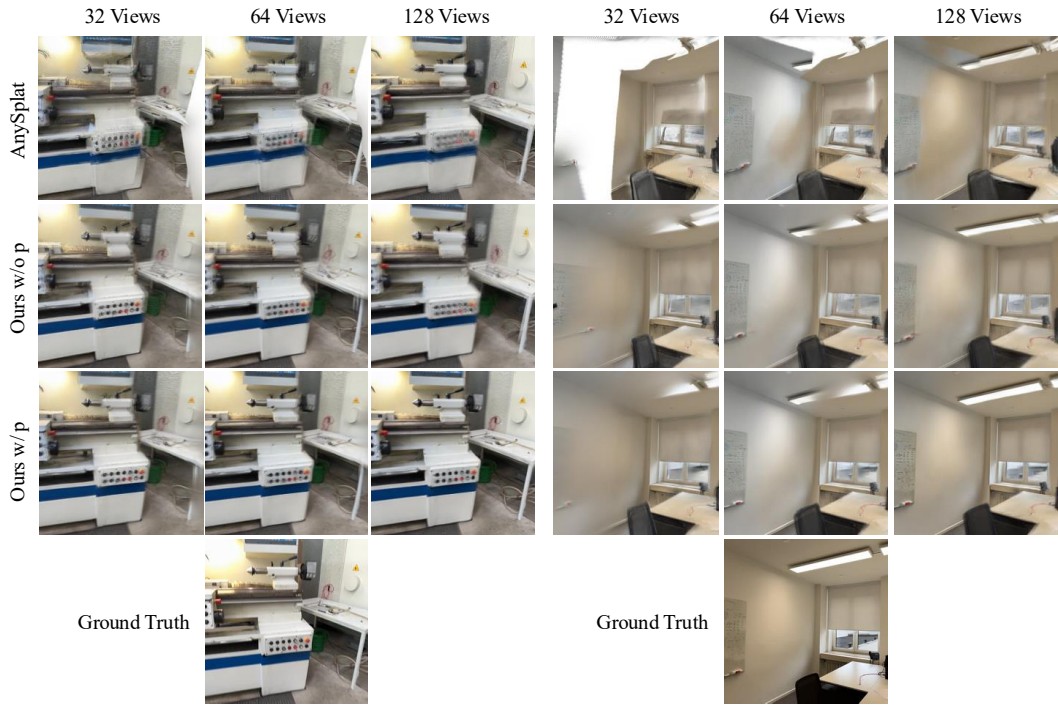

Figure 10: **More qualitative comparison on ScanNet++.** YoNoSplat demonstrates strong generalization to the unseen ScanNet++ dataset, producing more coherent reconstructions than AnySplat by better fusing multi-view Gaussians. The quality improves as more input views are provided (left to right). Notably, while our model performs well without priors (Ours w/o p.), providing ground-truth intrinsics (Ours w/ p.) further enhances generalization and fusion, leading to the highest fidelity results.

stantially higher than those produced by standard calibration pipelines, where intrinsic errors are typically much smaller. Therefore, our conditioning remains robust in realistic scenarios.

## C  LIMITATIONS

Our method leverages a feedforward approach to reconstruct wide-coverage scenes from an arbitrary number of unposed images. However, the maximum number of input views is constrained by GPU memory. Therefore, an interesting future direction is to explore incremental feedforward reconstruction (Wang et al., 2025b). Moreover, as shown in Tab. 1, pose optimization can still substantially improve the performance of our Gaussians, indicating that the current feedforward model has significant potential for further enhancement. We also observe that drastic illumination changes between views (e.g., day vs. night) can break photometric consistency, leading to geometric inaccuracies and floating artifacts. Future work could address this issue by explicitly training on datasets with diverse illumination conditions.

## D  MORE VISUAL COMPARISONS

Here, we provide more qualitative comparisons on the ScanNet++ (Yeshwanth et al., 2023), RealEstate10K (Zhou et al., 2018), and DL3DV (Ling et al., 2024) datasets. As shown in Fig. 10, Fig. 11, and Fig. 12, our pose-free method consistently outperforms the previous SOTA pose-free method, NoPoSplat (Ye et al., 2025). Moreover, we can even achieve superior novel view rendering quality compared to SOTA pose-required methods (Chen et al., 2024; Xu et al., 2025) and optimization-based methods (Fan et al., 2024).

# E    USE OF LARGE LANGUAGE MODELS

We use LLMs solely for improving grammar, wording, and overall readability of the manuscript. The model is not used for ideation, experimental design, implementation, or analysis. All technical content, methodology, and results are original and developed entirely by the authors.

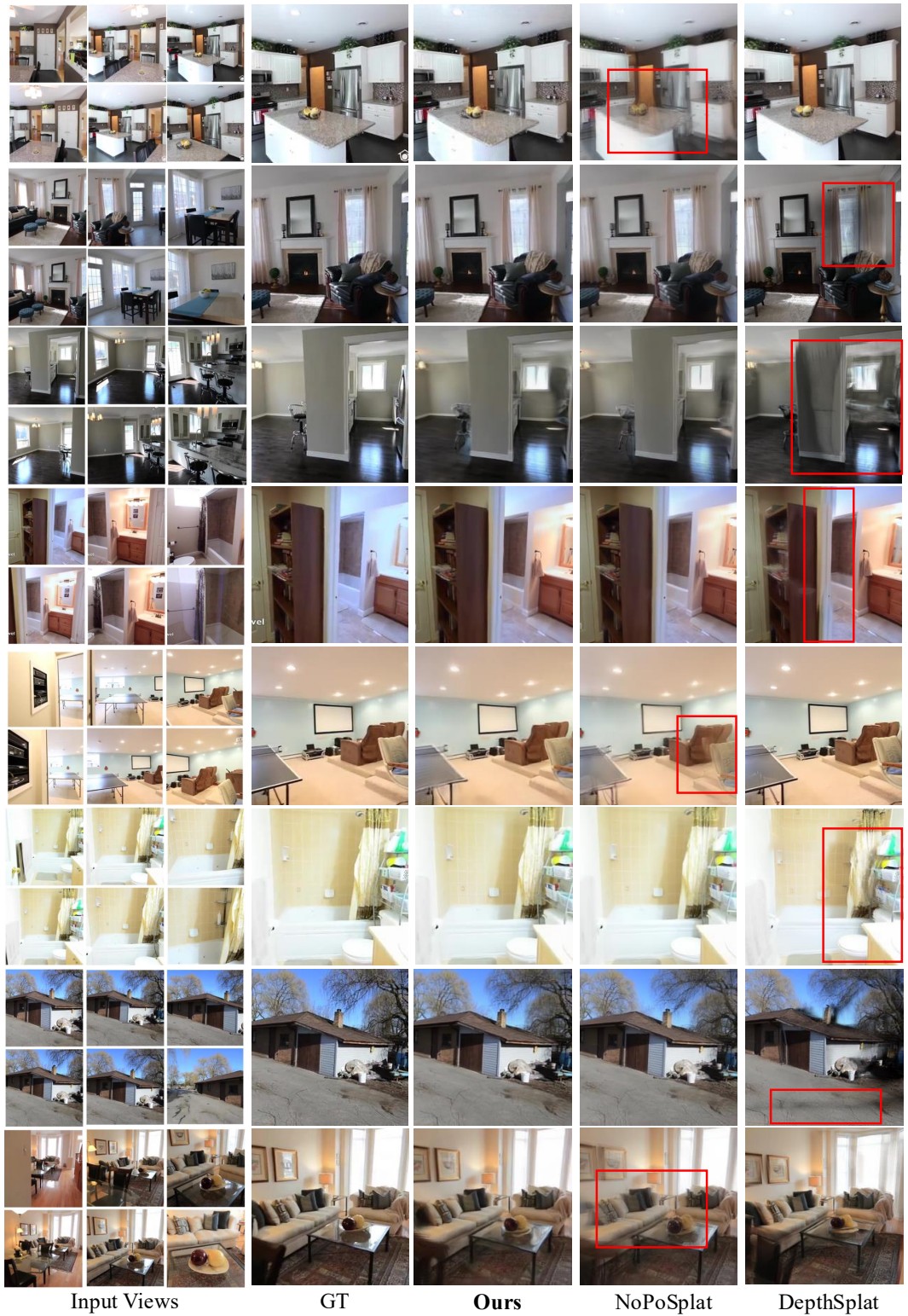

|  |  |  |  |  |
|---|---|---|---|---|
| Input Views | GT | **Ours** | NoPoSplat | DepthSplat |

Figure 11: **Qualitative comparison on RealEstate10K (Zhou et al., 2018)**. Our method achieves high-quality novel view synthesis compared with the previous SOTA pose-required method (Xu et al., 2025) and pose-free method (Ye et al., 2025).

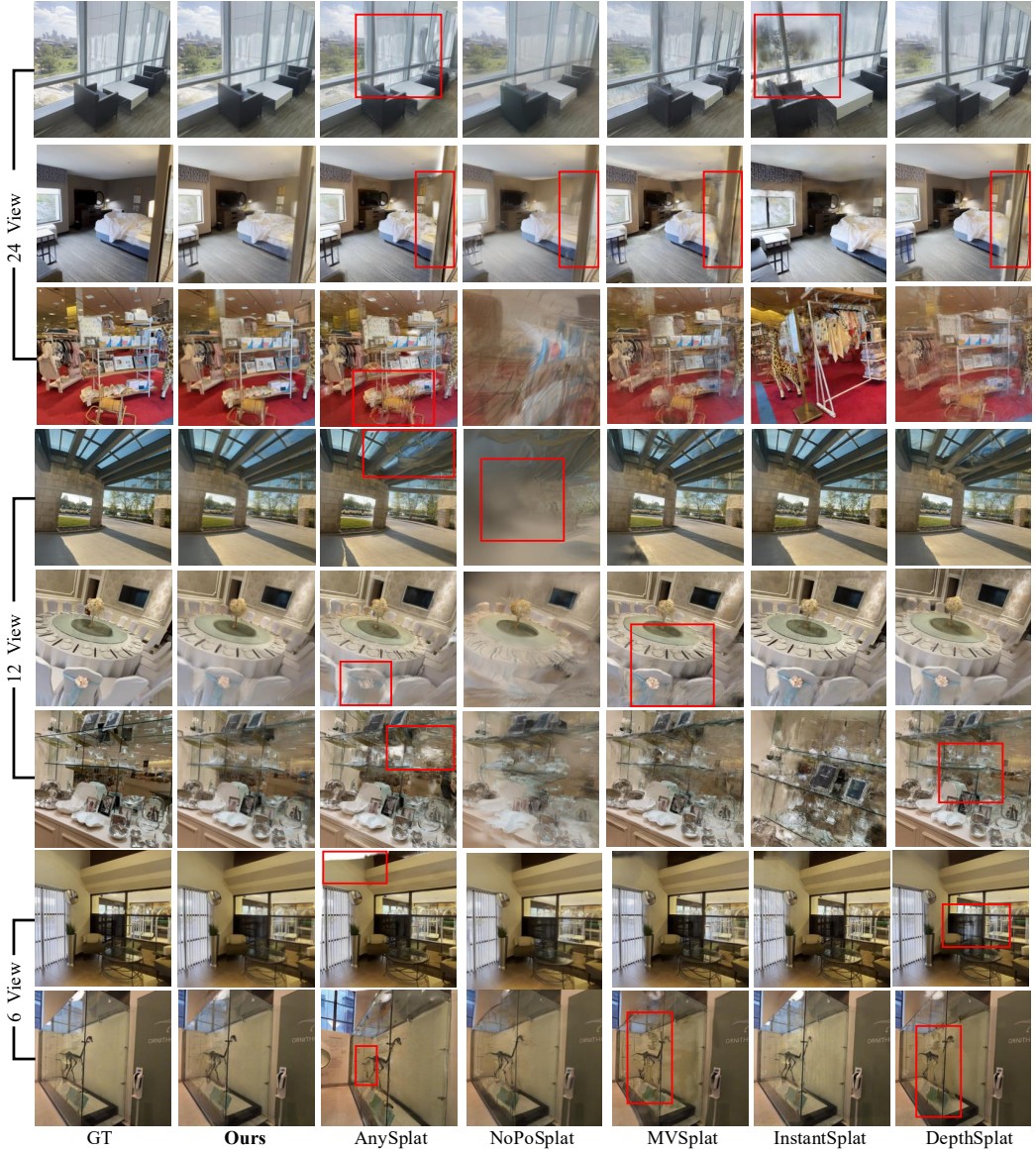

Figure 12: **Qualitative comparison on DL3DV (Ling et al., 2024)**. We compare our method with state-of-the-art optimization-based (Fan et al., 2024), pose-required (Chen et al., 2024; Xu et al., 2025), and pose-free (Ye et al., 2025; Jiang et al., 2025) methods. Here, the results of our method are obtained under the *pose-free, intrinsic-free* setting. The results demonstrate that our method generates novel view images of higher quality than these methods.

