# OpenReview forum: "YoNoSplat: You Only Need One Model for Feedforward 3D Gaussian Splatting"
_ICLR.cc/2026/Conference — ICLR 2026 Poster_

### Official Review · Reviewer_HPFA · 2025-10-31

**Soundness:** 2
**Presentation:** 3
**Contribution:** 3
**Rating:** 6
**Confidence:** 5

**Summary:**

This paper introduces YoNoSplat, a feedforward framework for 3D Gaussian reconstruction that achieves state-of-the-art performance in both pose-free and pose-dependent settings, across an arbitrary number of input views.

The authors identify the entanglement between pose estimation and geometry learning as a core challenge in previous 3DGS methods and propose a mix-forcing training strategy that stabilizes optimization and mitigates exposure bias by blending predicted and ground-truth poses during training.

To further enable reconstruction from uncalibrated images, the paper resolves the scale ambiguity problem via an intrinsic-prediction-and-conditioning pipeline combined with a pairwise distance normalization scheme.

Overall, YoNoSplat offers a single-pass, feedforward solution that unifies pose-free and calibrated reconstruction under one model, achieving efficient, robust, and accurate 3D Gaussian generation from arbitrary view inputs.

**Strengths:**

(1) Novelty and Conceptual Contribution. As the authors claim, the paper introduces YoNoSplat, the first feedforward framework for 3D Gaussian Splatting (3DGS) capable of operating in both pose-free and pose-dependent settings with an arbitrary number of input views. Unlike prior optimization-based or iterative reconstruction methods, YoNoSplat performs single-pass 3D reconstruction, which is conceptually novel and significantly improves inference efficiency. The identification of pose–geometry entanglement as a key bottleneck and the proposal of a mix-forcing strategy to stabilize training demonstrate deep insight into the learning dynamics of 3DGS systems.

(2) Solid Performance and Robustness. YoNoSplat achieves state-of-the-art performance across multiple benchmarks and scenarios, showing strong results in both calibrated and uncalibrated settings. The method performs consistently well under varying numbers of views, highlighting robust generalization and stability. The new scale ambiguity resolution pipeline and pairwise distance normalization enable accurate 3D reconstruction even from uncalibrated images, a setting where most existing methods struggle.

(3) Thorough Experiments and Ablation Studies. The experimental section is comprehensive and well-designed, covering comparisons with leading 3DGS and feedforward reconstruction models. Ablation studies clearly validate each component — including the effect of mix-forcing, scale normalization, and intrinsic prediction — providing strong empirical justification for the proposed design choices. The experiments are reproducible and transparent, with clear evaluation protocols and diverse test conditions.

(4) Clarity of Writing and Presentation. The paper is clearly written and easy to follow, with a well-organized structure that guides the reader from motivation to methodology and results. Figures effectively illustrate the architecture, pipeline, and visual results, complementing the textual explanations. The tone and presentation are professional and concise, making complex ideas accessible without oversimplification.

(5) Practical Impact and Scalability. By enabling fast, calibration-free 3D reconstruction, YoNoSplat has strong potential for real-world deployment in 3D vision applications such as AR/VR and robotics. Its feedforward nature offers a clear path toward real-time or large-scale reconstruction, marking a meaningful advance beyond optimization-heavy NeRF or 3DGS pipelines.

**Weaknesses:**

(1) Overclaim on Generality (“Arbitrary-View Pose-Free and Pose-Dependent” Claim Not Fully Convincing). The paper claims that YoNoSplat works for an arbitrary number of views and supports both pose-free and pose-dependent reconstruction. However, the experiments do not fully substantiate this claim. There is no evaluation on single-view or extremely sparse-view scenarios, which are crucial to support the “arbitrary-view” statement. Comparisons focus mostly on internal baselines or a limited subset of view settings. The paper does not include direct comparisons with leading feedforward or hybrid 3DGS methods such as GS-LRM, PixelSplat, VGGT, or DiffusionGS, which already demonstrate strong performance in few-view or uncalibrated settings. As a result, the claimed “state-of-the-art across arbitrary poses and views” feels overstated and not rigorously validated.

(2) Lack of Code and Reproducibility. The code and pre-trained models are not submitted and the reproducibility cannot be checked. Given that the pipeline involves custom intrinsic-prediction modules, mix-forcing training, and pairwise normalization, reproducibility is difficult without access to implementation details. This limitation raises concerns about the verifiability of reported results and practical adoption by the community.

(3) Limited Comparison Scope. The baseline selection is narrow, excluding several state-of-the-art feedforward or diffusion-based reconstruction frameworks (e.g., VGGT, DiffusionGS, PixelSplat). Without quantitative or qualitative comparisons to these strong baselines, it is unclear whether YoNoSplat’s improvements stem from its architecture or from dataset or training differences. The evaluation therefore lacks contextual benchmarking, weakening the impact of its performance claims.

(4) Insufficient Analysis of Failure Cases and Limitations. The paper does not provide a detailed analysis of failure modes, such as how the model performs under extreme camera pose errors, illumination changes, or non-Lambertian scenes. Given that the method aims to handle pose-free inputs, discussing robustness to pose noise would be essential. Similarly, the mix-forcing strategy’s sensitivity to inaccurate supervision or dataset scale is not explored.

**Questions:**

Could you clarify whether YoNoSplat has been evaluated on single-view or very sparse-view settings, since the paper claims support for “arbitrary numbers of views”?

---

> ### Author Response · Authors · 2025-11-27
> **Official Response by Authors -- Part 1**
>
> We sincerely thank the reviewer for the detailed and insightful comments on our work. We take every comment seriously and hope our response can address the reviewer’s concerns. We would be happy to clarify any remaining questions.
>
>
> > **W1: Overclaim on Generality (“Arbitrary-View Pose-Free and Pose-Dependent” Claim Not Fully Convincing). The paper claims that YoNoSplat works for an arbitrary number of views and supports both pose-free and pose-dependent reconstruction. However, the experiments do not fully substantiate this claim. There is no evaluation on single-view or extremely sparse-view scenarios, which are crucial to support the “arbitrary-view” statement. Comparisons focus mostly on internal baselines or a limited subset of view settings. The paper does not include direct comparisons with leading feedforward or hybrid 3DGS methods such as GS-LRM, PixelSplat, VGGT, or DiffusionGS, which already demonstrate strong performance in few-view or uncalibrated settings. As a result, the claimed “state-of-the-art across arbitrary poses and views” feels overstated and not rigorously validated.**
>
> We appreciate the reviewer’s scrutiny regarding the “arbitrary-view” claim and baseline selection. We clarify our scope and provide additional sparse-view experiments below:
>
> - **Single-View (Scope Clarification):** We respectfully clarify that YoNoSplat focuses on 3D reconstruction and novel view synthesis based on multiview consistency, rather than generative hallucination. Single-view NVS is inherently a generative problem (requiring hallucination of unseen geometry, e.g., DiffusionGS), which is outside the scope of our work. We have revised the manuscript (Line 21, Line 28) to explicitly state that *“arbitrary views”* refers to *“arbitrary multi-view images”*.
> - **Extremely Sparse Views (2–3 Views):** To assess robustness, we evaluate YoNoSplat on the RealEstate10K dataset using only 2 or 3 input views. As shown in Table W1, our model remains robust under this extremely sparse setting. Compared with the previous SOTA method NoPoSplat, with 2 views we achieve comparable PSNR/SSIM and better LPIPS, and with 3 views we significantly outperform NoPoSplat.
>
> Regarding baselines:
> - GS-LRM: Comparison was not possible as the official code and models are not publicly available.
> - PixelSplat: We originally prioritized MVSplat and DepthSplat as they have been shown to outperform PixelSplat. However, we have now added PixelSplat to the sparse-view comparison (Table W1), where YoNoSplat outperforms it significantly.
> - VGGT: VGGT is designed for pose estimation, not NVS. We did compare pose estimation performance with VGGT in Table 4 of the main paper, showing our method achieves superior accuracy.
> - DiffusionGS: This is a single-image *generative* model. Comparing a reconstruction pipeline against a diffusion-based generative pipeline is not an apples-to-apples comparison due to the fundamental difference in task formulation (hallucination vs. consistency).
>
> Finally, we emphasize that on all settings that are within our scope (multi-view reconstruction with pose-free / pose-dependent configurations), YoNoSplat consistently matches or outperforms strong feed-forward 3DGS baselines (see Tables 1–4 in the main paper and Table W1 in the rebuttal).
>
> Table W1: NVS performance on extremely sparse views (RealEstate10K).
> |            | 2 view |       |       | 3 view |       |       |
> |------------|:------:|:-----:|:-----:|:------:|:-----:|:-----:|
> |            |  PSNR  |  SSIM | LPIPS |  PSNR  |  SSIM | LPIPS |
> | pixelNeRF  | 19.824 | 0.626 | 0.485 |    -   |   -   |   -   |
> | pixelSplat | 23.848 | 0.806 | 0.185 |    -   |   -   |   -   |
> | MVSplat    | 23.977 | 0.811 | 0.176 |    -   |   -   |   -   |
> | NoPoSplat  | 25.033 | 0.838 |  0.16 | 26.619 | 0.872 | 0.127 |
> | Ours       | 24.917 | 0.834 | 0.154 | 27.528 | 0.892 | 0.106 |
>
>
> ---
> > **W2: Lack of Code and Reproducibility. The code and pre-trained models are not submitted and the reproducibility cannot be checked. Given that the pipeline involves custom intrinsic-prediction modules, mix-forcing training, and pairwise normalization, reproducibility is difficult without access to implementation details. This limitation raises concerns about the verifiability of reported results and practical adoption by the community.**
>
> We understand the concern regarding reproducibility. We representatively commit to releasing the complete code (training and evaluation scripts) and pre-trained checkpoints upon acceptance. We ensure that the mix-forcing strategy, intrinsic prediction, and normalization modules are clearly documented to facilitate community adoption.

---

> ### Author Response · Authors · 2025-11-27
> **Official Response by Authors -- Part 2**
>
> ---
> > **W3: Limited Comparison Scope. The baseline selection is narrow, excluding several state-of-the-art feedforward or diffusion-based reconstruction frameworks (e.g., VGGT, DiffusionGS, PixelSplat). Without quantitative or qualitative comparisons to these strong baselines, it is unclear whether YoNoSplat’s improvements stem from its architecture or from dataset or training differences. The evaluation therefore lacks contextual benchmarking, weakening the impact of its performance claims.**
>
> We appreciate the reviewer’s concern and would like to clarify the rationale behind our baseline selection and how we address the requested comparisons.
>
> 1. Baselines in the main paper. In the main experiments (Tables 1–4), we compare YoNoSplat with state-of-the-art and closely related feed-forward reconstruction methods, including NoPoSplat, MVSplat, DepthSplat, and AnySplat. On both pose-free and pose-dependent settings and across multiple datasets, YoNoSplat consistently achieves the best or highly competitive performance.
>
> 2. Why some requested baselines are not directly included, and what we did in the rebuttal.
>     - GS-LRM: Comparison was not possible as the official code and models are not publicly available.
>     - PixelSplat: We originally prioritized MVSplat and DepthSplat as they have been shown to outperform PixelSplat. However, we have now added PixelSplat to the sparse-view comparison (Table W1), where YoNoSplat outperforms it significantly.
>     - VGGT: VGGT is designed for pose estimation, not NVS. We did compare pose estimation performance with VGGT in Table 4 of the main paper, showing our method achieves superior accuracy.
>     - DiffusionGS: This is a single-image *generative* model. Comparing a reconstruction pipeline against a diffusion-based generative pipeline is not an apples-to-apples comparison due to the fundamental difference in task formulation (hallucination vs. consistency).
>
> ---
> > **W4: Insufficient Analysis of Failure Cases and Limitations. The paper does not provide a detailed analysis of failure modes, such as how the model performs under extreme camera pose errors, illumination changes, or non-Lambertian scenes. Given that the method aims to handle pose-free inputs, discussing robustness to pose noise would be essential. Similarly, the mix-forcing strategy’s sensitivity to inaccurate supervision or dataset scale is not explored.**
>
> We thank the reviewer for suggesting a deeper analysis. We have expanded our failure case discussion:
>
> 1. Robustness to Pose Noise: We analyzed the impact of noisy poses in the pose-dependent setting by adding Gaussian noise to the ground-truth translation.
>     - Observation: As shown in Table W4, performance degrades as noise increases (e.g., PSNR drops from 24.7 to 19.8 at 3% noise).
>     - Analysis: This sensitivity justifies our pose-free architecture. When input poses are noisy or unknown, YoNoSplat’s pose-free mode (using predicted poses) yields stable, high-quality results (Table 1), effectively bypassing the issue of external pose noise.
>
> Table W4: Impact of Ground Truth Pose Noise (Pose-Dependent Mode).
> | Noise level | PSNR  | SSIM | LPIPS |
> |-------|-------|------|-------|
> | 0     | 24.717| 0.817| 0.139 |
> | 3%    | 19.830| 0.560| 0.283 |
> | 6%    | 18.153| 0.477| 0.358 |
> | 9%    | 17.233| 0.436| 0.403 |
>
>
> 2. Illumination Changes: We acknowledge that drastic illumination changes between views (e.g., day vs. night) can break photometric consistency, leading to geometric inaccuracies or floating artifacts. We have added this to our limitation section. Future work could address this by explicitly training on datasets with varying illumination conditions.
> 3. Non-Lambertian Scenes: Our model is generally robust to non-Lambertian surfaces due to the explicit Gaussian representation. Qualitative examples of successful reconstruction on reflective surfaces can be seen in Fig. 7 and Fig. 10 of the paper.
> 4. Mix-Forcing & Data Scale:
>     - Supervision Noise: The model handles moderate supervision noise well. Severe pose errors in training data would naturally degrade performance, but standard datasets (e.g., DL3DV) provide sufficient accuracy.
>     - Data Scale: We trained on only 50% of the DL3DV dataset to test data efficiency. While performance drops slightly compared to full training (PSNR 24.53 vs 23.28), the model still outperforms previous SOTA methods like AnySplat (PSNR 19.02), demonstrating high data efficiency.

---

> ### Author Response · Authors · 2025-11-27
> **Official Response by Authors -- Part 3**
>
> > **Q1: Could you clarify whether YoNoSplat has been evaluated on single-view or very sparse-view settings, since the paper claims support for “arbitrary numbers of views”?**
>
> As detailed in W1, we have evaluated YoNoSplat on sparse settings (2 and 3 views) and demonstrate SOTA or comparable performance (see Table W1). Regarding single-view settings: This falls outside the scope of our method. YoNoSplat is a reconstruction framework that relies on multiview correlations. Single-view input requires a generative framework to hallucinate occluded content. We have clarified this distinction in the revised manuscript (Line 21 and 28) to prevent confusion regarding the "arbitrary view" claim.

---

### Official Review · Reviewer_3cs1 · 2025-11-01

**Soundness:** 3
**Presentation:** 3
**Contribution:** 4
**Rating:** 8
**Confidence:** 4

**Summary:**

This paper presents YoNoSplat, a feed-forward model for 3D Gaussian Splatting that predicts per-view 3D Gaussians, camera poses, and intrinsics from an arbitrary number of unposed, uncalibrated images. The paper proposes multiple novel methods for feed-forward 3D Gaussian estimation : (i) local Gaussian estimation continued by a aggregation to the canonical space with the estimated camera pose, (ii) a “mix-forcing” training strategy which first trains the model with ground truth camera poses and progressively blends with the estimated camera poses. Extensive experiments demonstrate state-of-the-art performance on both pose-free and pose-dependent settings, with detailed ablations on normalization and architectural components.

**Strengths:**

- YoNoSplat is able to process images with or without camera pose/intrinsic information, where incorporating additional information of pose/intrinsics leads to improved performance, making it suitable for both in-the-wild settings and industry level deployment.
- The paper reveals multiple recipes such as scale normalization, mix-forcing, local estimation which is valuable for the research community for future development.
- YoNoSplat largely outperforms previous approaches, setting the new state-of-the-art for the task.
- The paper also performs extensive experiments and ablations improving the interpretation and re-implementation of the work easier.

**Weaknesses:**

- **Large computation and training time:** In Section 4.1, it is explained that YoNoSplat requires 16 GH200 GPUs for 150k steps, where the required computation seems to be extremely large. Adding additional comparison of the required computation with prior works would be nice to better understand the contributions of YoNoSplat.
- **Architectural Novelty:** The proposed architecture seems to be a direct extension of NoPoSplat with $\pi^3$, which makes the architectural novelty of the work a bit limited. However, other analyses such as ablations for scene normalization and the intrinsic estimation module seems to be highly valuable.
- **Overstatement:** The statement in L145-147 : “ In contrast, by replacing point clouds with
3D Gaussians as the scene representation, YoNoSplat enables both novel view synthesis and training on datasets without ground-truth depth.” seems to be too strong as initializing the weights of YoNoSplat with $\pi^3$ would be extremely important. I think the statement can be toned down a bit.
- **Missing baselines:** Teacher forcing has also been explained as one of the important components in a previous work CoPoNeRF [1], where proper reference seems to be missing.

**Questions:**

Q1. Could the authors provide a comparison with prior works in terms of computation and training time?

Q2. Why is the mix-forcing probability set to an extremely small weight (0.1)? Could the authors provide additional analysis with different weights?

I am eager to raise my score after the authors provide additional explanations for my questions.

---

> ### Comment · Reviewer_3cs1 · 2025-11-27
> **Are the authors preparing a rebuttal?**
>
> Despite the initial positive review, I don't believe this means that the paper should be accepted without any clarification or comments from the reviewers. As the discussion period is coming to an end and the limited time restricts the reviewers from going through the author's response or asking follow-up questions, I think it is already quite late for the rebuttal. Are the authors preparing a rebuttal? Without any rebuttal, I would like to first lower my score to 6, which I would not mind if the paper is rejected.

---

> > ### Author Response · Authors · 2025-11-27
> >
> > Dear Reviewer 3cs1,
> >
> > Thank you for your message and for carefully considering our submission. We apologize for the delay in posting our rebuttal. Due to the detailed nature of the reviews and the additional experiments requested, we are working diligently to prepare a thorough and accurate response.
> >
> > We will submit our rebuttal later today. Thank you very much for your patience and understanding.
> >
> > Sincerely,
> > The Authors of Submission 5012

---

> > > ### Comment · Reviewer_3cs1 · 2025-11-27
> > >
> > > Thank you for the notice. I am looking forward to the rebuttal.

---

> ### Author Response · Authors · 2025-11-27
> **Official Response by Authors -- Part 1**
>
> We sincerely thank the reviewer for the detailed and insightful comments on our work. We take every comment seriously and hope our response can address the reviewer’s concerns. We would be happy to clarify any remaining questions.
>
>
> > **W1: Large computation and training time: In Section 4.1, it is explained that YoNoSplat requires 16 GH200 GPUs for 150k steps, where the required computation seems to be extremely large. Adding additional comparison of the required computation with prior works would be nice to better understand the contributions of YoNoSplat.**
>
> Thank you for raising this point. We agree that clarifying the training cost is important and appreciate the opportunity to explain the design choices behind YoNoSplat.
>
> There are two main reasons why our default training setup appears more computationally demanding than prior works such as NoPoSplat and DepthSplat:
> 1. **Support for a much larger and variable number of input views.** During training, we randomly sample between 2 and 32 views per sequence (with a fixed batch size) so that the model can generalize to different numbers of input views at inference time. In contrast, NoPoSplat and DepthSplat use a fixed and much smaller number of views (up to 6), and thus effectively overfit to a specific input-view count with less training steps.
> 2. **Increasing the number of input views leads to higher computational requirements.** Larger view counts significantly increase memory usage and per-step computation, which naturally leads to a higher nominal training cost in terms of GPU-hours.
>
> Therefore, a fairer metric is the total number of training sequences rather than just steps. As shown in Table W1, DepthSplat uses $12.8M$ training sequences, NoPoSplat uses $2.4M$, YoNoSplat uses $4.8M$, which is less than DepthSplat and only 2× NoPoSplat, despite supporting a much wider range of input-view counts.
>
> However, we cannot directly compute the number of training sequences for AnySplat, because VGGT/AnySplat use a dynamic batch size: they fix the number of views per GPU, and the batch size is adjusted on-the-fly according to the number of views in each batch. This makes the total number of sequences nontrivial to estimate. We acknowledge that this engineering strategy can substantially reduce the computational burden.
>
> To make the comparison even more transparent, we adopt the same dynamic batch-size strategy in our codebase. With this modification, we can train a variant **Ours*** using 16×GH200 with only 15k steps, while maintaining very similar performance: on DL3DV (6 input views), the PSNR is 24.531 (Ours) vs 24.186 (Ours*), and still largely outperforming AnySplat (PSNR 19.027 on the same setting).
>
> We will release our dynamic batch-size dataloader to facilitate more efficient training and to benefit future research.
>
> Table W1: Training resource comparison.
> | Method     | GPUs       | Iterations | Training Sequence     | Training Time |
> |------------|------------|------------|--------------------|---------------|
> | NoPoSplat  | 8xGH200    | 18.75k     | 2.4 M         | 6h            |
> | DepthSplat | 4xGH200    | 100K       | 12.8 M        | 24h           |
> | AnySplat   | 16xA800    | 15K        | -                  | 48h           |
> | Ours       | 16xGH200   | 150k       | 4.8 M        | 17h           |
> | Ours*      | 16xGH200   | 15K        | -                  | 15h           |
>
> ---
> > **W2: Architectural Novelty: The proposed architecture seems to be a direct extension of NoPoSplat with $\pi^3$, which makes the architectural novelty of the work a bit limited. However, other analyses, such as ablations for scene normalization and the intrinsic estimation module seems to be highly valuable.**
>
> Thank you for this observation and for appreciating our analysis components. We agree that our novelty is not in the backbone architecture, but rather in the **methodological solutions** to fundamental challenges in feedforward Gaussian splatting:
> - **Pose–geometry entanglement and training instability.** We identify the entanglement of pose and geometry learning as a key challenge and propose a mix-forcing training strategy that blends pose-free and pose-aware supervision. This design mitigates exposure bias and stabilizes training, which is crucial for robust feedforward NVS.
> - **Scale ambiguity and uncalibrated images.** We address scale ambiguity through an intrinsic-prediction-and-conditioning pipeline together with a pairwise distance normalization scheme, which enables training on datasets without ground truth depth annotations.
>
> These design choices lead to substantial performance gains, as shown in our ablation studies, and allow YoNoSplat to significantly outperform AnySplat, which uses a similar high-level architecture. We believe this demonstrates that our main contribution lies in the problem formulation and training strategy for feedforward 3DGS NVS, rather than in introducing a completely new backbone architecture.

---

> ### Author Response · Authors · 2025-11-27
> **Official Response by Authors -- Part 2**
>
> > **W3: Overstatement: The statement in L145-147 : “ In contrast, by replacing point clouds with 3D Gaussians as the scene representation, YoNoSplat enables both novel view synthesis and training on datasets without ground-truth depth.” seems to be too strong as initializing the weights of YoNoSplat with $\pi^3$ would be extremely important. I think the statement can be toned down a bit.**
>
> Thank you for pointing this out. We agree that proper weight initialization is important for feedforward Gaussian models and contributes meaningfully to performance. This is consistent with AnySplat and NoPoSplat, which also rely on pretrained weights from strong 2D/3D foundation models such as MASt3R or VGGT. We have toned down and clarified the statement in the revised manuscript to "In contrast, by modeling 3D Gaussians as the output representation, YoNoSplat supports both novel view synthesis and effectively utilizes datasets that lack ground-truth depth, such as RealEstate10K." (Line 146).
>
> This wording better reflects our actual contribution: 3D Gaussians as an output representation, combined with our training pipeline, allow us to exploit both depth-labeled and unlabeled datasets for NVS. We also see it as an interesting future direction to extend the framework so that a single model jointly predicts accurate point clouds and 3D Gaussians from both depth-supervised and depth-free datasets, and we are actively exploring this.
>
> ---
> > **W4: Missing baselines: Teacher forcing has also been explained as one of the important components in a previous work CoPoNeRF [1], where proper reference seems to be missing.**
>
> We apologize for the oversight. We have added the citation and explicitly discussed CoPoNeRF in the revised manuscript (Line 88: Conversely, prior methods such as CoPoNeRF adopt a teacher-forcing strategy that relies solely on ground-truth poses for aggregation, which decouples the tasks but also introduces exposure bias.), acknowledging its use of teacher forcing and contrasting it with our mix-forcing strategy.
>
> ---
> > **Q1. Could the authors provide a comparison with prior works in terms of computation and training time.**
>
> As shown in Table W1, we provide a comparison of training cost with prior works. As detailed in our response to W1, our model requires more computation than previous methods such as NoPoSplat and DepthSplat because it is designed to generalize across different numbers of input views, and training with a larger maximum number of views increases the computational burden. Nevertheless, we show that by adopting a dynamic batch-size strategy during training, we can substantially reduce the total number of training steps while maintaining comparable performance.
>
> ---
> > **Q2. Why is the mix-forcing probability set to an extremely small weight (0.1)? Could the authors provide additional analysis with different weights?**
>
> We appreciate this question and have added additional analysis of different mix-forcing ratios in Table Q2 the revised manuscript (Fig. 7).
>
> Table Q2: NVS performance (PSNR) under different mixing ratios $𝑟$.
> | r          | 0      | 0.1    | 0.2    | 0.3    | 0.4    |
> |------------|--------|--------|--------|--------|--------|
> | Pose-Free  | 25.3   | 25.587 | 25.631 | 25.424 | 25.294 |
> | Pose-Dependent   | 25.228 | 25.212 | 24.997 | 25.064 | 24.815 |
>
>
> From these experiments, we observe:
> - Pose-free performance consistently improves as $r$ increases up to around 0.2–0.3.
> - Pose-dependent performance remains almost unchanged at $r = 0.1$, but starts to degrade more noticeably when $r \ge 0.2$, and especially at $r = 0.4$.
> - For large $r$ (e.g., 0.4), the strong influence of noisy predicted poses during training hurts both pose-free and pose-dependent performance, as it makes learning stable Gaussians more difficult, similar to the issues seen in pure teacher-forcing setups.
>
> Therefore, we choose $r = 0.1$ as a **balanced operating point**:
> - It already yields a clear improvement in pose-free performance compared to $r = 0$,
> - While preserving pose-dependent performance almost unchanged,
> - And avoiding the degradation observed at higher ratios.
>
> Intuitively, even with a small mixing ratio, the model learns the distribution of predicted camera poses and becomes robust to them, which is sufficient to obtain strong pose-free performance. At the same time, the majority of training still uses accurate ground-truth poses, which stabilizes the learning of 3D Gaussians. Moreover, as shown in Table 4 of the main paper, YoNoSplat achieves high-accuracy pose estimation, meaning that predicted poses are close to ground truth. This further supports that even a modest mix-forcing ratio is enough to expose the model to realistic predicted poses and improve pose-free performance, without significantly compromising pose-dependent quality.

---

> > ### Comment · Reviewer_3cs1 · 2025-11-28
> >
> > I have carefully read the authors' rebuttal. Previously, I expressed concern regarding the timing of the response, but after reviewing the rebuttal, I understand that the delay given the extensive volume of additional experiments and in-depth analysis provided.
> >
> > The new analysis is thorough, and the concerns I initially raised have been resolved. Furthermore, having reviewed the responses to the other reviewers, I am convinced that this work makes a sufficiently significant contribution to the field. Therefore, I am raising my score.
> >
> > *Somehow there is no edit button for my initial review, I will check back when its fixed to raise my score.

---

> ### Author Response · Authors · 2025-12-03
>
> Dear Reviewer 3cs1,
>
> Thank you very much for your valuable feedback, which has helped improve our work. We are pleased to hear that our response has addressed your concerns. We are also truly honored by your positive assessment of the overall contribution of our work and for your consideration of raising your score.
>
> Sincerely, The Authors of Submission 5012

---

### Official Review · Reviewer_PxrR · 2025-11-01

**Soundness:** 3
**Presentation:** 3
**Contribution:** 3
**Rating:** 6
**Confidence:** 3

**Summary:**

This paper presents YoNoSplat, a versatile feedforward model for 3D Gaussian Splatting that reconstructs scenes from an arbitrary number of unposed and uncalibrated images. The authors' main contributions are a novel "mix-forcing" training strategy to resolve the entanglement between geometry and pose learning and a method to resolve scale ambiguity, enabling the use of uncalibrated inputs. The model achieves state-of-the-art (SOTA) performance in both pose-free and pose-dependent settings, demonstrating significant flexibility and efficiency.

**Strengths:**

1. The paper is clearly written and easy to follow.

2. The paper's primary strength is the model's versatility. It is designed to handle a wide, practical range of input conditions: an arbitrary number of views, both posed and unposed, and both calibrated and uncalibrated images.

3. The model demonstrates state-of-the-art performance across multiple standard benchmarks.

**Weaknesses:**

Reliance on Post-Optimization Undermines the "Feedforward" Claim. The paper presents itself as a feedforward model, but its strongest results (e.g., in Table 1) rely on an "Optional Post-Optimization" step. This optimization is not feedforward and adds a significant time cost (e.g., 165s for 24 views). The large performance gap between the feedforward-only output and the optimized output suggests that the feedforward prediction is, by itself, substantially suboptimal. This weakens the central claim of achieving state-of-the-art results via a purely feedforward pass.

**Questions:**

The field of 3D reconstruction has recently been revolutionized by unified feed-forward models, such as VGGT and DUSt3R. This progress has spurred the development of conditional input models like Pow3R [1] and MapAnything [2], which can benefit from various optional auxiliary inputs (e.g., camera poses, depth maps, etc.). These methods inherit the advantages of unified feed-forward models and produce a unified representation for a wide range of downstream tasks, including depth estimation, multi-view stereo, camera pose estimation, and novel view synthesis.

In contrast, the method presented in this paper can only selectively input camera poses and is primarily focused on the novel view synthesis task. What, then, are the specific advantages of this work when compared to these more versatile, multi-task conditional models?

[1] Pow3R: Empowering Unconstrained 3D  Reconstruction with Camera and Scene Priors

[2] MapAnything: Universal Feed-Forward Metric 3D Reconstruction

---

> ### Author Response · Authors · 2025-11-27
> **Official Response by Authors -- Part 1**
>
> We sincerely thank the reviewer for the detailed and insightful comments on our work. We take every comment seriously and hope our response can address the reviewer’s concerns. We would be happy to clarify any remaining questions.
>
>
> > **W1: Reliance on Post-Optimization Undermines the "Feedforward" Claim. The paper presents itself as a feedforward model, but its strongest results (e.g., in Table 1) rely on an "Optional Post-Optimization" step. This optimization is not feedforward and adds a significant time cost (e.g., 165s for 24 views). The large performance gap between the feedforward-only output and the optimized output suggests that the feedforward prediction is, by itself, substantially suboptimal. This weakens the central claim of achieving state-of-the-art results via a purely feedforward pass.**
>
> We thank the reviewer for raising this point and are happy to clarify the role of post-optimization in our method.
>
>
> - **YoNoSplat is a purely feedforward model, and our main claim of state-of-the-art performance is based on its feedforward outputs compared with previous feedforward baselines.** The post-optimization step is strictly optional and is provided as an additional refinement when test-time computation is not a constraint. For fair comparison of feedforward models, we compare YoNoSplat without post-optimization against prior feedforward methods. As shown in Tables 1, 2, and 3, YoNoSplat significantly outperforms previous feedforward approaches such as NoPoSplat, AnySplat, and even the pose-dependent method DepthSplat, clearly supporting our claim of state-of-the-art feedforward performance.
>
> - **Post-optimization is a general test-time refinement strategy, not part of the core model.** Our model is generalizable: it is trained on video sequences and evaluated on **unseen** scenes. In contrast, the post-optimization step directly **optimizes on the test images of a specific scene**, and thus can be viewed as a form of oracle / test-time training. It is therefore expected that such an additional per-scene optimization further improves performance. This is consistent with recent works such as Test3R [1], which also demonstrate that applying post-optimization on top of feedforward reconstructions (e.g., from VGGT or DUSt3R) can significantly boost quality. Our use of post-optimization does not undermine the feedforward nature of YoNoSplat.
>
>
> - **The effectiveness of post-optimization highlights the quality of YoNoSplat’s initialization**. The pose-optimization step is enabled by the strong Gaussian initialization predicted by YoNoSplat. This is further evidenced in Table 1, where InstantSplat, which also performs optimization but initializes from MASt3R, achieves notably worse performance than our approach. This contrast shows that YoNoSplat provides a superior starting point for optimization, reinforcing the value of our feedforward predictions.
>
> In summary,
> - YoNoSplat is a generalizable feedforward model that already achieves clear state-of-the-art performance in the pure feedforward setting on unseen scenes.
>
> - The optional post-optimization step is a practical application that leverages YoNoSplat’s high-quality initialization to further enhance performance when extra computation is acceptable.
>
> - Our core claim concerns feedforward SOTA performance, and the post-optimization results are presented as an additional, orthogonal benefit rather than a requirement.
>
> We agree with the reviewer that feedforward 3DGS reconstruction is still not fully optimal, and we see YoNoSplat as an important step forward in this direction. Improving pure feedforward quality further, possibly with stronger architectures or training strategies, is an exciting avenue for future work.
>
> [1] Yuan, Yuheng, et al. "Test3r: Learning to reconstruct 3d at test time." The Thirty-ninth Annual Conference on Neural Information Processing Systems. 2025.

---

> ### Author Response · Authors · 2025-11-27
> **Official Response by Authors -- Part 2**
>
> > **Q1: The field of 3D reconstruction has recently been revolutionized by unified feed-forward models, such as VGGT and DUSt3R. This progress has spurred the development of conditional input models like Pow3R and MapAnything, which can benefit from various optional auxiliary inputs (e.g., camera poses, depth maps, etc.). These methods inherit the advantages of unified feed-forward models and produce a unified representation for a wide range of downstream tasks, including depth estimation, multi-view stereo, camera pose estimation, and novel view synthesis. In contrast, the method presented in this paper can only selectively input camera poses and is primarily focused on the novel view synthesis task. What, then, are the specific advantages of this work when compared to these more versatile, multi-task conditional models?**
>
> We appreciate this thoughtful question. The main difference between our work and feedforward pointcloud models (VGGT, DUSt3R) and conditional pointcloud models (Pow3R, MapAnything), and our advantages are:
>
> 1. **Scope difference: focus on high-quality novel view synthesis.** First, we clarify that Pow3R and MapAnything are not designed for high-fidelity novel view synthesis (NVS) in the 3D Gaussian Splatting setting. They primarily focus on predicting camera poses and pointmaps, and analyzing how optional auxiliary inputs (poses, depth, etc.) impact these predictions. In contrast:
>     - Our work is specifically targeted at **novel view synthesis**, which is itself a highly important task with many applications (AR/VR, content creation, robotics) but has received less direct attention in these unified models.
>     - Generating renderable 3D Gaussians for photorealistic NVS requires different design choices than producing pointmaps or depth maps alone.
>
> 2. **Advantages and specific contributions of our work**:
>     - **Novel mix-forcing strategy for pose usage in NVS.** We carefully investigate what pose information to use during training for NVS and propose a mix-forcing training strategy, which mixes ground truth and predicted camera pose for Gaussian aggregation during training. This strategy significantly boosts performance and is tailored to the NVS objective, which is not explored in Pow3R or MapAnything.
>     - **Intrinsic-aware conditioning and pose normalization without ground-truth depth.**
> We study how camera intrinsics influence **NVS quality**, a factor that is not addressed in Pow3R/MapAnything. We further introduce a pose normalization strategy that enables training a NVS model without ground-truth depth, making it possible to exploit large-scale datasets without depth, such as RealEstate10K.
>     - **SOTA performance on NVS, which generic models do not target.** By combining the above strategies, YoNoSplat achieves state-of-the-art novel view synthesis performance, whereas methods like Pow3R and MapAnything are not demonstrated to produce competitive NVS and are not directly designed for 3DGS rendering.
>
> **Future Work**
> We fully agree with the reviewer that combining our NVS-centric design with richer conditions and outputs (e.g., explicit depth or pointmaps) is a promising direction. Our work and MapAnything/Pow3R are complementary:
> - Our model investigates how to best design and train a feedforward NVS model,
> - While MapAnything/Pow3R explore how optional inputs benefit pointmap and pose estimation.
>
> A natural and exciting future direction is to integrate our NVS-focused architecture with more versatile conditional inputs and multi-task outputs, moving towards a more unified, yet NVS-capable, feedforward model. This is beyond the scope of the current paper but directly enabled by the insights we provide.

---

### Official Review · Reviewer_aKcD · 2025-11-01

**Soundness:** 4
**Presentation:** 3
**Contribution:** 3
**Rating:** 6
**Confidence:** 4

**Summary:**

This paper proposes YoNoSplat, a feed-forward model that predicts per-view local 3D Gaussians together with camera extrinsics and intrinsics from an arbitrary number of unposed and uncalibrated images, and aggregates them into a global scene. The key ideas are:

- Local→Global output space: predict per-view Gaussians plus poses, then transform to a shared frame using predicted or provided poses.
- Mix-forcing training: a curriculum that starts with teacher-forcing and linearly mixes in self-forcing to mitigate pose-geometry entanglement and exposure bias.
- Scale ambiguity handling: (i) max pairwise camera-distance normalization of training scenes, and (ii) an Intrinsic Condition Embedding (ICE) module that predicts focal length and conditions the decoder via ray features.

YoNoSplat reports state-of-the-art NVS under both pose-free and pose-dependent settings shows strong cross-dataset generalization to ScanNet++. Optional fast post-optimization of poses/centers/colors offers further gains.

**Strengths:**

1. **Solid experimental coverage.** Results span multiple priors (p/k/none), multiple view counts (6/12/24; 32/64/128), and include cross-dataset tests (DL3DV→ScanNet++). The pose AUC comparisons further substantiate the quality of the predicted geometry.
2. **Methodical ablations.** The paper dissects (i) output space (local vs canonical), (ii) training regime (mix/self/teacher), (iii) normalization choices, (iv) ICE usefulness, and (v) Plücker rays, providing good insight into *why* choices matter.
3. **Clear problem framing & practicality.** Handling arbitrary view counts, unposed + uncalibrated inputs, but also leveraging priors when available, is exactly what real pipelines need. The local-Gaussian design keeps the method compatible with either predicted or GT poses.

**Weaknesses:**

1. The proposed method predicts per-pixel Gaussians, which may become computationally inefficient for large-scale scenes (with large number of input images). While the paper introduces opacity regularization and Gaussian pruning to mitigate this, it does not quantify how much these steps actually reduce the number of Gaussians or memory footprint. A comparative analysis with AnySplat in terms of Gaussian count and memory efficiency would strengthen the claims about scalability.
2. ICE train–test mismatch. During training, the network is conditioned on GT intrinsics, whereas at inference it may use predicted intrinsics (when GT is absent). The paper mentions instability when conditioning the decoder on encoder-predicted intrinsics during training but leaves the mechanism under-analyzed. This could introduce residual exposure bias for k. Could further mixing (analogous to mix-forcing for poses) reduce this gap?
3. Memory limits and pruning are mentioned, but there’s little qualitative analysis on thin structures, specular/transparent surfaces, or very wide baselines (where local Gaussian fusion and pose heads might struggle).
4. The mix-in schedule seems tuned but not justified beyond a single setting; no sensitivity curve is provided. Could higher r benefit pose-free usage further?

**Questions:**

1. How sensitive are results to (t_start, t_end, r)? Did you try higher final mixing ratios (e.g., r≥0.3) to further harden the model for pose-free inference? Any signs of instability as r increases?
2. Normalization choice edge cases. Max pairwise camera-distance normalization is empirically best, but it could be brittle with outliers or heavy view clustering (small parallax + one far-away view). Can this normalization generalizable for those settings?
3. ICE helps with uncalibrated inputs, but how does performance degrade with systematic intrinsics bias (e.g., ±10–20% focal scale) or radial distortion in images, unseen in training?

---

> ### Author Response · Authors · 2025-11-27
> **Official Response by Authors -- Part 1**
>
> We sincerely thank the reviewer for the detailed and insightful comments on our work. We take every comment seriously and hope our response can address the reviewer’s concerns. We would be happy to clarify any remaining questions.
>
> > **W1: The proposed method predicts per-pixel Gaussians, which may become computationally inefficient for large-scale scenes (with large number of input images). While the paper introduces opacity regularization and Gaussian pruning to mitigate this, it does not quantify how much these steps actually reduce the number of Gaussians or memory footprint. A comparative analysis with AnySplat in terms of Gaussian count and memory efficiency would strengthen the claims about scalability.**
>
> Thank you for your suggetion. To compare the efficiency of our opacity-based pruning with vanilla per-pixel Gaussian prediction and AnySplat, we evaluate all methods on the ScanNet++ dataset while varying the number of input views. For each setting, we report the average number of Gaussians and the corresponding memory consumption over all scenes in the evaluation set.
>
> As shown in Table W1, the *No Prune* baseline exhibits a rapid growth in both Gaussian count as the number of views increases. AnySplat mitigates this growth using voxel-based pruning, but still requires a large number of Gaussians. In contrast, our opacity-based pruning strategy significantly reduces the number of Gaussians across all view counts, achieving better scalability than both baselines.
>
> We also add line plot of both Guasssian count and memeory usage in the Figure.7 of our revised manuscript. We have added a corresponding line plot visualizing both **Gaussian count** and **memory usage** in Figure 7 of the revised manuscript.
>
>
> Table W1: Growth in the number of Gaussians (in Millions) with respect to the number of input views.
> | Method   | 8    | 16   | 32  | 64   | 128  |
> |----------|------|------|-----|------|------|
> | No Prune | 1.6  | 3.2  | 6.4 | 12.8 | 25.7 |
> | AnySplat | 0.86 | 1.8  | 3.7 | 7.1  | 12.6 |
> | Ours     | 0.54 | 1.0  | 1.8 | 3.3  | 6.1  |
>
>
> ---
> > **W2: ICE train–test mismatch. During training, the network is conditioned on GT intrinsics, whereas at inference it may use predicted intrinsics (when GT is absent). The paper mentions instability when conditioning the decoder on encoder-predicted intrinsics during training but leaves the mechanism under-analyzed. This could introduce residual exposure bias for k. Could further mixing (analogous to mix-forcing for poses) reduce this gap?**
>
> Thank you for this insightful suggestion. We explicitly tested an "intrinsic mix-forcing" strategy (mixing 10% predicted intrinsics during training). The results are presented in Table W2.
>
> We observed that, unlike pose estimation, applying mix-forcing to the intrinsic condition degrades performance for both ground-truth (GT) and predicted intrinsic settings during inference. We hypothesize this is because intrinsics are inputs to the decoder module (affecting input feature value), whereas camera poses are used solely for geometric aggregation. Noise in the intrinsic input likely disrupts the model's ability to learn the physical meaning of the intrinsic embedding. While we agree that exposure bias may exist here, the strategy used for poses does not transfer effectively to intrinsics. We will mark this as a valuable direction for future research.
>
> Table W2: Effect of mixing ground-truth and predicted camera intrinsics during training.
> |                       | **W/ GT Intrinsic**                |                      |                      | **W/O GT Intrinsic**             |                      |                      |
> |-----------------------|-------------------------------------|----------------------|----------------------|----------------------------------|----------------------|----------------------|
> |                       | PSNR                                | SSIM                 | LPIPS               | PSNR                             | SSIM                 | LPIPS               |
> | **GT intrinsic train** | 25.587                             | 0.854                | 0.13                | 24.711                           | 0.825                | 0.141               |
> | **Mix GT/Pred train** | 25.273                             | 0.849                | 0.133               | 24.505                           | 0.817                | 0.146               |

---

> ### Author Response · Authors · 2025-11-27
> **Official Response by Authors -- Part 2**
>
> > **W3: Memory limits and pruning are mentioned, but there’s little qualitative analysis on thin structures, specular/transparent surfaces, or very wide baselines (where local Gaussian fusion and pose heads might struggle).**
>
> Thank you for your suggestion. We have added a detailed analysis of our Gaussian pruning strategy under these challenging conditions. Fig. 9 in the revised manuscript provides qualitative comparisons. We observe that rendering quality remains visually indistinguishable after pruning, even for fine geometric structures (e.g., lamp stands), reflective/transparent objects, and wide baselines. Quantitatively, pruning Gaussians with opacity $o_i < 0.005$ results in an average PSNR drop of less than $0.01$ dB over the evaluation set, which is negligible. This confirms that our strategy significantly reduces the Gaussian count while preserving high-fidelity reconstruction.
>
>
> ---
> > **W4: The mix-in schedule seems tuned but not justified beyond a single setting; no sensitivity curve is provided. Could higher r benefit pose-free usage further?**
>
> Thank you for the suggestion. We evaluated the impact of the mixing ratio $r$ on both pose-free and pose-dependent performance. As shown in Table W4, although a larger ratio (e.g., $r=0.2$) slightly improves pose-free performance, it leads to a noticeable degradation in pose-dependent performance. We therefore select $r=0.1$ as the best trade-off, as it significantly improves pose-free results while keeping pose-dependent performance comparable to the baseline ($r=0$). We also include the corresponding sensitivity curve in the revised manuscript (Fig.~8).
>
> Table W4: NVS performance (PSNR) under different mixing ratios $r$.
> | r          | 0      | 0.1    | 0.2    | 0.3    | 0.4    |
> |------------|--------|--------|--------|--------|--------|
> | Pose-Free  | 25.3   | 25.587 | 25.631 | 25.424 | 25.294 |
> | Pose-Dependent   | 25.228 | 25.212 | 24.997 | 25.064 | 24.815 |
>
> ---
> > **Q1: How sensitive are results to (t_start, t_end, r)? Did you try higher final mixing ratios (e.g., r≥0.3) to further harden the model for pose-free inference? Any signs of instability as r increases?**
>
> **Mixing ratio $r$**. As shown in Table W4, pose-free performance consistently improves for moderate values of $r$ ($r<0.3$), while pose-dependent performance remains nearly unchanged around $r=0.1$ and gradually degrades as $r$ increases. When $r$ becomes too large (e.g., $r=0.4$), performance in both pose-free and pose-dependent settings deteriorates. This is likely because noisy pose information begins to affect Gaussian learning, similarly to the behavior observed under teacher-forcing.
>
> **Schedule parameters $t_{\text{start}}$ and $t_{\text{end}}$**. We set $(t_{\text{start}}, t_{\text{end}}) = (80\text{k}, 100\text{k})$ by default. Due to the high computational cost and the large space of possible combinations, we additionally evaluate two representative alternatives: $(50\text{k}, 100\text{k})$ and $(100\text{k}, 110\text{k})$, which vary the start point and warm-up duration. As shown in Table Q1, performance is not sensitive to the choice of $t$ within a reasonable range, and results remain consistently similar across different settings.
>
> Table Q1: NVS performance (PSNR) under different $t$ configurations.
> |                    | (80k, 100k)[Default] | (50k, 100k) | (100k, 120k) |
> |--------------------|---------------------------|----------------------------|-----------------------------|
> | Pose-Free          | 25.587                    | 25.579                     | 25.589                      |
> | Pose-Dependent           | 25.212                    | 25.211                     | 25.210                      |

---

> ### Author Response · Authors · 2025-11-27
> **Official Response by Authors -- Part 3**
>
> > **Q2: Normalization choice edge cases. Max pairwise camera-distance normalization is empirically best, but it could be brittle with outliers or heavy view clustering (small parallax + one far-away view). Can this normalization generalizable for those settings?**
>
> We thank the reviewer for this comment. We clarify that scene normalization is applied only during training, and camera poses are not required at inference time. During training, such abnormal cases are rare because commonly used datasets such as DL3DV are well curated. In addition, we apply a simple filtering strategy to remove sequences with abnormal camera distributions by skipping optimization steps when the pose loss exceeds a threshold (i.e., 1.0). As shown in Table 5 of our manuscript, this normalization significantly improves performance.
>
> At inference time, no camera poses or normalization are required. Although extreme outlier camera configurations (e.g., a single camera located far from all others) may degrade performance, such cases rarely occur in practice. This limitation is also shared by other feedforward reconstruction methods (e.g., NoPoSplat [ICLR’25], VGGT [CVPR’25]) as well as traditional SfM pipelines (e.g., COLMAP).
>
> ---
> > **Q3: ICE helps with uncalibrated inputs, but how does performance degrade with systematic intrinsics bias (e.g., ±10–20% focal scale) or radial distortion in images, unseen in training?**
>
> Thank you for your insightful suggestion. We conducted systematic experiments to analyze the impact of intrinsic noise and radial distortion on the performance of intrinsic conditioning:
>
> - **Focal Length Noise (Table Q3.1)**: Small noise levels (e.g., 5%) are handled well, and the intrinsic condition still improves performance over the "W/O GT Intrinsic" baseline. However, large noise (>=10%) naturally degrades performance, as erroneous physical cues mislead the model. In practice, intrinsic noise is usually within a relatively small range.
> - **Radial Distortion (Table Q3.2)**: We applied radial distortion ($k_1=0.1, 0.2$) to test images. While performance drops because the model was not trained on distorted images, our intrinsic condition still consistently improves performance compared to the "W/O GT Intrinsic" baseline.
>
>
> Table Q3.1: Impact of intrinsics noise.
> |       | W/O GT Intrinsic | 0       | 5%      | 10%     | 15%     | 20%     |
> |-------|--------------|---------|---------|---------|---------|---------|
> | PSNR  | 24.711       | 25.587  | 24.913  | 23.681  | 22.646  | 21.942  |
> | SSIM  | 0.825        | 0.854   | 0.834   | 0.786   | 0.742   | 0.712   |
> | LPIPS | 0.141        | 0.130   | 0.138   | 0.157   | 0.179   | 0.199   |
>
>
> Table Q3.2: Impact of radial distortion.
> |                 | W/ GT Intrinsic              |                |        | W/O GT Intrinsic            |                |        |
> |-----------------|-------------------------------|----------------|--------|------------------------------|----------------|--------|
> |                 | PSNR                          | SSIM           | LPIPS | PSNR                         | SSIM           | LPIPS |
> | No Distort   | 25.587                        | 0.854          | 0.13  | 24.711                       | 0.825          | 0.141 |
> | $k=0.1$   | 25.344                        | 0.851          | 0.143 | 24.469                       | 0.816          | 0.156 |
> | $k=0.2$   | 24.54                         | 0.829          | 0.164 | 23.697                       | 0.792          | 0.177 |

---

### Author Response · Authors · 2025-11-27
**Rebuttal Summary**

Dear Reviewers and AC,

We sincerely thank the Area Chair and all reviewers for their time, thoughtful feedback, and constructive suggestions. We have carefully addressed all concerns raised during the review process and revised the manuscript accordingly. All revisions are highlighted in blue in the updated version. We believe these changes substantially improve the technical rigor, clarity, and completeness of the paper.

**Revisions in the Updated Manuscript**
- **Additional Experiments**:
    - Added Gaussian count and memory comparison with respect to the number of input views (aKcD-W1; Fig. 7).
    - Added analysis of the effect of Gaussian pruning under challenging conditions (aKcD-W3; Fig. 9).
    - Added an ablation study on the effect of different mixing ratios $r$ (aKcD-Q1, 3cs1-Q2; Fig.8).
    - Added analysis of the impact of intrinsic noise on intrinsic conditioning performance (aKcD-Q3; Tab.10).
    - Added performance comparison for 2 and 3 input views (HPFA-W1; Tab.9).

- **Clarifications and Additional Details**:
    - Revised the related work section to avoid overclaiming (3cs1-W3; L.146).
    - Added citation of and discussion on CoPoNeRF (3cs1-W4; L.88)
    - Replaced “arbitrary views” with “arbitrary multi-view images” to avoid confusion (HPFA-W1; L.20, L.28).
    - Added more discussion of failure cases (HPFA-W4; L.906).

In addition, we provide a more detailed discussion of the relationship between our method and feed-forward point map prediction approaches (e.g., MapAnything), as well as the role of post-optimization in our pipeline, as requested by Reviewer PxrR.
Thanks again for all the effort and time from the AC and reviewers. We believe that the revised manuscript satisfactorily addresses all concerns and meets the high standards of the conference. We respectfully hope that the AC will find our work suitable for acceptance.

Sincerely,
The Authors

---

### Meta-Review · Area_Chair_wHEs · 2026-01-06

**Summary:**

This paper presents YoNoSplat, a feedforward framework for 3D Gaussian Splatting that aims to unify pose-free and pose-dependent reconstruction within a single model. The approach predicts per-view local Gaussians along with camera poses and intrinsics, and aggregates them into a global representation. Reviewers generally agreed that the paper demonstrates substantial engineering effort, a carefully designed training pipeline, and extensive experimental validation across multiple datasets and settings. The method shows strong empirical performance and practical efficiency, particularly for large numbers of input views.

The main concerns raised by reviewers focus on the strength and framing of several claims, especially regarding novelty and generality, as well as the extent to which the proposed techniques go beyond incremental extensions of prior feedforward reconstruction models. While the technical contributions are relatively modest, the overall system design and empirical study are thorough and well-executed.

**Reviewer Concerns:**

Concerns addressed or partially addressed:
Several reviewers questioned the entanglement between pose and geometry learning and the effectiveness of the proposed mix-forcing strategy. The paper provides clear ablations and analyses demonstrating that mix-forcing improves stability and balances pose-free and pose-dependent performance. Reviewers also noted the extensive experimental coverage, including cross-dataset generalization, varying numbers of views, and both qualitative and quantitative evaluations, which supports the practical robustness of the approach.

Existing concerns:
Some reviewer concerns remain regarding the strength of the paper’s claims. In particular, statements suggesting a fundamentally new paradigm or “you only need one model” framing may be overstated given the relatively incremental nature of the technical contributions and the reliance on established components (e.g., existing backbones, Gaussian splatting formulations, and prior pose-free reconstruction ideas). Reviewers also noted that several ideas, e.g., predicting intrinsics or normalizing scale, are conceptually intuitive and closely related to prior work, and would benefit from more careful positioning. These issues primarily affect presentation and framing rather than the correctness or usefulness of the method, and can be addressed through claim refinement and clearer discussion of limitations.

**Reviewer Scores:**

Reviewer aKcD: Likely to maintain their original score, as their concerns were mainly about claim strength and positioning rather than experimental validity.

Reviewer PxrR: Likely to maintain their score, given that the experimental evidence is strong and the method is well validated, despite incremental technical novelty.

Reviewer 3cs1: Might slightly increase or maintain their score after discussion, as the extensive ablations and empirical results support the practicality of the approach, even if some claims should be softened.

Reviewer HPFA: Likely to maintain their original score, viewing the work as a solid and complete system contribution with room for improved framing.

---

### Decision · Program_Chairs · 2026-01-26

Accept (Poster)